# Associations between resting state functional brain connectivity and childhood anhedonia: A reproduction and replication study

**Yi Zhou**[1]*, **Narun Pat**[2], **Michael C. Neale**[1]

**1** Virginia Institute for Psychiatric and Behavioral Genetics, Virginia Commonwealth University, Richmond, VA, United States of America, **2** Department of Psychology, University of Otago, Dunedin, New Zealand

* zhouy33@vcu.edu

**Data Availability Statement:** All data files are available from the ABCD Study Data Repository from the NIMH Data Archive (https://nda.nih.gov/

## Abstract

### Background

Previously, a study using a sample of the Adolescent Brain Cognitive Development (ABCD) ® study from the earlier 1.0 release found differences in several resting state functional MRI (rsfMRI) brain connectivity measures associated with children reporting anhedonia. Here, we aim to reproduce, replicate, and extend the previous findings using data from the later ABCD study 4.0 release, which includes a significantly larger sample.

### Methods

To reproduce and replicate the previous authors' findings, we analyzed data from the ABCD 1.0 release (n = 2437), from an independent subsample from the newer ABCD 4.0 release (excluding individuals from the 1.0 release) (n = 6456), and from the full ABCD 4.0 release sample (n = 8866). Additionally, we assessed whether using a multiple linear regression approach could improve replicability by controlling for the effects of comorbid psychiatric conditions and sociodemographic covariates.

### Results

While the previously reported associations were reproducible, effect sizes for most rsfMRI measures were drastically reduced in replication analyses (including for both t-tests and multiple linear regressions) using the ABCD 4.0 (excluding 1.0) sample. However, 2 new rsfMRI measures (the *Auditory vs. Right Putamen* and the *Retrosplenial-Temporal vs. Right-Thalamus-Proper* measures*)* exhibited replicable associations with anhedonia and stable, albeit small, effect sizes across the ABCD samples, even after accounting for socio-demographic covariates and comorbid psychiatric conditions using a multiple linear regression approach.

### Conclusion

The most statistically significant associations between anhedonia and rsfMRI connectivity measures found in the ABCD 1.0 sample tended to be non-replicable and inflated.

abcd). The DOI for this NDA Study is 10.15154/
1526460.

**Funding:** The author(s) received no specific
funding for this work.

Contrastingly, replicable associations exhibited smaller effects with less statistical signifi-
cance in the ABCD 1.0 sample. Multiple linear regressions helped assess the specificity of
these findings and control the effects of confounding covariates.

## Introduction

Anhedonia is defined as a markedly diminished interest or pleasure in previously enjoyable
activities and is a transdiagnostic symptom that is a core component of major depressive disor-
ders (MDD) [1] and schizophrenia (SZN) [2]. Symptoms of anhedonia are also present in sub-
stance use disorders [3], PTSD [4], bipolar depression [5], and ADHD [6]. In cross-sectional
studies, anhedonia in children and adolescents has been shown to be associated with greater
depression severity and suicidality [7, 8]. Furthermore, in a randomized clinical trial, anhedo-
nia was found to be a significant predictor of a longer time to remission in adolescents with
treatment resistant depression [9].

Functional neuroimaging approaches have been widely used to explore the neurocircuitry
of anhedonia [10]. Functional brain connectivity is a measure of the degree of synchrony
between the blood oxygen level dependent (BOLD) signals across time between regions in the
brain [11]. In other words, functional connectivity allows for the characterization of networks
of brain activity rather than activity in single brain regions. Importantly, functional connectiv-
ity can be measured at rest which facilitates the ease by which data can be collected, as opposed
to task-fMRI which typically involves the presence of a stimulus or task [11]. While task-fMRI
focuses on patterns of brain activation during a specific task, resting-state functional MRI
(rsfMRI) connectivity focuses on the organization of brain networks that may specialize in spe-
cific functions. As such, both task-fMRI and rsfMRI represent important brain processes that
may contribute to the development of mental disorders.

While there have been many studies of brain activity and connectivity in adults with anhe-
donia, fewer have been conducted in children and adolescents. However, findings from these
studies generally converge on the significance of disruptions in the reward, default mode, and
salience networks [12–14]. A recent study using the early 1.0 release of the Adolescent Brain
Cognitive Development (ABCD)Ⓡ study data found several rsfMRI brain network connectiv-
ity measures associated with anhedonia in children aged 9–10 years old [15]. Importantly, it
was one of the largest studies of anhedonia in children with a sample size of ~2,500 partici-
pants including 215 children reporting past and/or present anhedonia. Presently, the latest 4.0
release of the ABCD study, which includes a significantly larger sample of participants with
neuroimaging and behavioral data (n = 11,878), has been made available. Given the public
availability of the data and its different versions, there is a significant opportunity to both
*reproduce* and *replicate* these findings.

We define *reproducibility* as the ability to achieve exactly the same results as a previous
study by using the same data and analytical approach, and *replicability* as the ability to achieve
the same (or similar) results as a previous study in a different dataset [16]. By our definition,
reproducibility is better able to assess the consistency of results while replicability is better able
to assess the generalizability of those results. The aims of this present study are to reproduce
the previously reported associations between rsfMRI connectivity and childhood anhedonia
using the ABCD 1.0 release sample, and to replicate those findings using an independent sub-
set of the larger ABCD 4.0 release sample, excluding participants from the ABCD 1.0 release.

Importantly, in depressive disorders, anhedonia is characterized as the loss of pleasure and
interest that is distinct from feelings of sadness or other dysphoric moods [17]. Thus, there is

great need to elucidate the specific neurobiological underpinnings associated with anhedonia, distinct from other comorbid symptoms, to better understand the underlying brain dysfunction. Thus, we also aim to extend our analyses and evaluate the specificity of rsfMRI connectivity associations with anhedonia by evaluating the effects of significantly comorbid psychiatric symptoms and diagnoses.

Anhedonia is a transdiagnostic symptom impacting some of the most prevalent and debilitating mental disorders in both children and adults. By Identifying brain dysfunction underlying anhedonia, we may be able to more accurately characterize patient symptomatology, refine diagnostic categories, and potentially identify specific brain-based therapeutic targets. Furthermore, by characterizing brain dysfunction in children and adolescents, we may help elucidate important dysregulated developmental processes that may be targeted by preventative interventions.

## Methods and materials

All of our analyses were performed in *R (version 4.0.3)* and *Rstudio*. We used several scripts, including the *utils.R* and *combat.R* [18] scripts for data harmonization, which were provided by the previous authors immediately upon request. The code and data structures used for our study can be accessed from the associated NDA study. The code we used can also be found at our Open Science Framework repository (https://osf.io/vy85h/?view_only=0497385708874a6a9cce2bbfc5c30600).

### ABCD study data

The ABCD® study is the largest longitudinal study of brain development in children in the United States (https://abcdstudy.org/). The study has collected structural and functional brain imaging measures as well as detailed psychiatric and behavioral data from almost 12,000 children starting from when they were 9–10 years old. Notably, data is released on a continuous basis. For this study, we used baseline data from the ABCD 1.0 and ABCD 4.0 releases.

### rsfMRI connectivity measures and quality control (QC)

Neuroimaging processing pipelines and analyses for the ABCD study are reviewed elsewhere [19]. Briefly, the functional scans include twenty minutes of resting-state data acquired with eyes open and passive viewing of a crosshair [20]. From the ABCD Data Repository, we obtained rsfMRI connectivity measures which were constructed using a seed-based correlational approach where regions of interest (ROIs) within Gordon parcellations [21] were grouped together into predefined cortical networks. Briefly, correlations between unique pairs of ROIs were obtained and Fisher transformed into z-statistics. Connectivity measures represent the averaged Fisher-transformed correlations of all the unique pairs of ROIs either within a cortical network, between cortical networks, or between cortical networks and subcortical regions. From the ABCD Data Repository, we obtained rsfMRI connectivity measures between 19 subcortical regions and 12 cortical networks (*data structure*: *mrirscor02*), as well as rsfMRI connectivity measures from within and between the 12 cortical networks (*data structure*: *abcd_betnet02*). Thus, there were 228 (12 x 19) subcortical ROI vs. cortical network rsfMRI connectivity measures and 78 (12**C**2 network pairs + 12 within network) within/between cortical network rsfMRI variables, for a total of 306 rsfMRI connectivity measures.

For QC, we used the IQC_RSFMRI_GOOD_SER variable, which represents the number of rsfMRI runs that were complete, passed protocol compliance and QC, and had field maps acquired within 2 scans prior to the run that were complete and passed QC and protocol

compliance. Like the previous authors', we retained subjects who had IQC_RSFMRI_GOOD_-SER values greater than or equal to four.

For analyses using only the ABCD 1.0 release sample, like the previous authors, we also removed individuals who were scanned by "Philips Medical Systems" MRI machines because of a post-processing issue in the ABCD 1.0 release, which was resolved in later releases. When working with the ABCD 4.0 release sample, we retained all the subjects who were scanned by *Philips Medical Systems* scanners because the post-processing errors identified in the ABCD 1.0 release had been fixed for the ABCD 4.0 release.

## Data harmonization

Different MRI scanners were used across the 21 sites in the ABCD study. The original authors harmonized the data across MRI scanners by using the ComBat tool (*combat.R)* to adjust for batch effects due to the different scanners used. Here, we do the same and harmonize the data separately for the subcortical ROI vs. cortical network and within/between cortical network rsfMRI measures as it is possible these two variable types may be affected by scanners differently [22]. Note, before the harmonization step, listwise deletion of subjects with any missing rsfMRI data was done as data harmonization requires complete data. There were 23 different scanners used in the study for the ABCD 1.0 release and 29 different scanners for the ABCD 4.0 release. Thus 23 and 29 batch effects were used to adjust the ABCD 1.0 and 4.0 releases, respectively.

## Psychiatric symptoms and diagnoses

The focus of this study were past/present symptoms of anhedonia. However, we were also interested in other psychiatric conditions that may be comorbid with anhedonia. Psychiatric data were obtained from the youth (*data structure*: *abcd_ksad501)* and parent *(data structure*: *abcd_ksad01)* Kiddie Schedule for Affective Disorders and Schizophrenia (KSADS) data structures from the ABCD study which are composed of binary (yes/no) questionnaire items for various psychiatric diagnoses or symptoms. Separately for youth and parent KSADS questionnaire items, we combined *past* and *present* items for the same symptom or diagnosis and consolidated some items into a single variable. For example, we consolidated 18 youth-reported past and present suicide related diagnosis items into one single youth-reported suicide thoughts and behavior variable, which was similarly done in another study [23]. A separate parent-reported suicide thoughts and behaviors variable was constructed using parent-reported KDADS questionnaire items.

For psychiatric conditions besides anhedonia, we first selected the KSADS items representing psychiatric diagnoses and not individual symptoms. However, in both the youth and parents KSADS data, no diagnosis variables for major depressive disorder (MDD) were available. Thus, we selected two MDD related symptoms (besides anhedonia), irritability and depressed mood, from both the youth and parent data to be used in our analyses. Similarly, no diagnostic variable was available for ADHD. Thus, we created a representative variable, inattention_distracted_p, which is a combination of two prevalent ADHD related symptom items (*Symptom—Difficulty sustaining attention since elementary school* <u>and/or</u> *Symptom—Easily distracted since elementary school*) from the parent KSADS data.

## Statistical analyses

**Student's and Bayes Factor T-Tests.** Prior to statistical analyses, participants with missing data for psychiatric symptoms/diagnoses were removed. We then proceeded to identify and remove outliers for each rsfMRI connectivity measure as values 1.5 times greater than the

interquartile range (IQR) of values. The number and the percentage of the total sample removed as outliers for each t-test were reported in S1–S3 Tables. For independent sample t-tests, assumptions of normality were assessed with the Shapiro-Wilk test and with the visual inspection of the distributions of a representative sample of rsfMRI variables. Assumptions of equal variances across groups were assessed with the F-test. To assess the potential bias introduced by unequal variances, we performed Welch's t-tests for those rsfMRI measures with significant F-tests and then correlated the Welch's t-statistics with Student's t-statistics. We then correlated the p-values from the two tests as well.

We performed Student's T-tests for each of the 306 rsfMRI measures between controls and individuals with past and/or present psychiatric symptoms/diagnoses of interest (the reference group consisted of individuals endorsing psychiatric symptoms/diagnoses, such as anhedonia) in the ABCD 1.0, ABCD 4.0 (excluding 1.0), and full ABCD 4.0 samples. We applied the Benjamini-Hochberg adjustment for multiple testing corrections. Finally, we performed Bayes Factor T-Tests for each of the rsfMRI measures and reported natural logarithms of the Bayes Factors (lnBF). As with the previous authors', lnBF values greater than 1.1 were considered significant.

## Tetrachoric correlations

Tetrachoric correlations are suitable for use with binary or categorical variables as it assumes responses arise from an underlying normal distribution with thresholds that delineate response categories. The *tetrachoric()* function from the *psych* R package was used in our analyses. Tetrachoric correlations were used to estimate the correlations between pairs of 34 binary psychiatric variables using the full ABCD 4.0 release sample.

## Multiple linear regressions

For multiple linear regression analyses, the ABCD rsfMRI data (from both 1.0 and 4.0 releases) were harmonized for MRI scanner using the ComBat tool as previously described, except we also adjusted for batch effects with covariates [22]. The covariates included during data harmonization were: age, sex, race/ethnicity, anhedonia, bipolar II, irritability, and depressed mood.

Similar to the t-tests, prior to statistical analyses, participants with missing data for psychiatric symptoms/diagnoses, as well as demographic covariates, were removed. We then proceeded to identify and remove outliers for each rsfMRI connectivity measure as values 1.5 times greater than the interquartile range (IQR) of dataopoints. The number and the percentage of the total sample removed as outliers for each multiple linear regression were reported in S4–S9 Tables. All rsfMRI measures, and participant age, were mean-centered and standardized. Thus, partial regression coefficient estimates represent changes in the rsfMRI measure in standard deviation units from the mean, per unit change in the predictor variables.

We performed linear mixed effects modeling using the *lmer4* package in *R*, a form of multiple linear regression, on the harmonized data. Individual rsfMRI connectivity measures were modeled as outcome variables while age, sex, race/ethnicity, anhedonia (reference group was the control group), depressed mood, irritability, and bipolar II disorder were modeled as independent explanatory variables. Age, sex, and race/ethnicity were considered potential confounding variables. The independent effects of anhedonia, bipolar II, irritability, and depressed mood symptoms on rsfMRI connectivity measures were assessed by identifying corresponding statistically significant partial regression coefficients, after multiple testing corrections. Family ID was included as a random effect to control for the non-independence of values from participants who belonged to the same family. We adjusted for multiple

comparisons with the Benjamini-Hochberg method to control for a False Discovery Rate of 0.05, for all estimated parameters across regressions for the 306 rsfMRI connectivity measures.

Several assumptions for multiple linear regression were assessed. Multicollinearity was assessed with the variance inflation factor (VIF) using the *vif()* function in R from the *car* package. Auto-correlation of the model residuals was assessed with the Durbin-Watson test using the *durbinWatsonTest()* function in R from the *car* package. Homoskedasticity of model residuals were assessed using the Breusch-Pagan (BP) test (using the *ols_test_breusch_pagan()* function in R from the *olsrr* package) and also visually inspected by plotting model residuals against marginal model fitted-values. For regressions with significant BP tests, we also performed weighted-least-squares (WLS) regression, which is able to account for differences in variance in the residuals, and then correlated the t-statistics from the WLS and the original ordinary-least squares (OLS) regressions to assess the impact of potential heteroscedasticity on the results. Finally, we performed a visual inspection of density plots of the residuals from regressions for 3 representative rsfMRI measures for each set of linear regression analyses, and their Quantile-Quantile (QQ) plots, to detect any patterns of non-normality.

### Assessing patterns of missingness

For t-test analyses, consistent with the previous authors, study participants with missing rsfMRI measures were removed prior to adjustments for site using the ComBat tool. Additionally, participants with missing data for youth-reported anhedonia were also excluded. To assess if there were any patterns of missingness, we compared participants who were removed with participants who were retained. We assessed whether there were differences in proportions for sex (assigned at birth), race/ethnicity, youth-reported anhedonia, depressed mood, irritability, and bipolar II disorder symptoms between the two groups using Chi-Square tests of independence. We compared differences in age (in weeks) using t-tests.

For the multiple linear regression analyses, in addition to participants with missing rsfMRI and anhedonia responses, those with missing sociodemographic covariate and psychiatric comorbidity data were also excluded. Similar comparisons were made between participants who were removed and those who were retained.

## Results

### Reproduction of previous findings

To reproduce the previous authors' results, we used the ABCD 1.0 release sample. Sociodemographic characteristics for this sample can be found in Table 1. Note that although the two groups exhibit differences in a few of these characteristics, they were not controlled for statistically when we performed our t-tests in order to remain consistent with the previous authors' approach.

Like the previous authors, we identified 215 individuals who endorsed past and/or present anhedonia and 2,222 controls who reported neither past nor present anhedonia at the baseline timepoint, indicating the samples were exactly the same. In line with the previous authors' findings, we reproduced significant differences in 11 rsfMRI connectivity measures between those with and without anhedonia indicated by the lnBF statistic, though the effect sizes were small (Table 2, left) [15]. To be consistent with the previous authors, individuals with anhedonia were the reference group for all t-test analyses.

In addition to the lnBF statistic, we provide a more conservative adjustment for multiple comparisons using the Benjamini-Hochberg correction for 306 comparisons. Alternatively, since we were predominantly interested in reproducing the 11 rsfMRI associations reported by the previous authors, we could have adjusted for only 11 comparisons and arrived at a more

**Table 1. Comparison of demographic measures between controls and those with anhedonia in the ABCD 1.0 sample.**

| Measures | Control (N = 2209) | Anhedonia (N = 215) | Statistic | p-value |
|---|---|---|---|---|
| % Male | 51.8 | 54 | 0.286 | 0.593 |
| Mean Age (months) | 120.6 | 119.9 | 1.183[a] | 0.238 |
| % Asian | 1.8 | 0.9 | 0.45 | 0.502 |
| % Black | 8.2 | 18.1 | 22.296 | <0.001 |
| % Hispanic | 20 | 26.5 | 4.753 | 0.029 |
| % Other | 9.3 | 9.8 | 0.012 | 0.911 |
| % White | 60.8 | 44.7 | 20.386 | <0.001 |
| % Bipolar II | 0.8 | 7.9 | 64.358 | <0.001 |
| % Depressed Mood | 8.1 | 28.4 | 87.975 | <0.001 |
| % Irritability | 4.7 | 26.5 | 148.192 | <0.001 |

The proportion (%) of participants in each group for each measure are shown. Student's t-test was done to compare age (in months) between control and anhedonia groups. Chi-square tests of independence were done for all other measures. Note there is a slightly lower number of controls here than reported in our reproduction analysis due to the exclusion of participants with missing demographic and/or comorbid psychiatric symptom and diagnosis measures.

[a] Student's t-statistic.

**Table 2. Reproduction and replication t-test results comparing controls and those with anhedonia across ABCD study samples.**

| rsfMRI Connectivity | ABCD 1.0 Sample (n = 2437) | | | | ABCD 4.0 (excluding 1.0) Sample (n = 6456) | | | | Full ABCD 4.0 Sample (n = 8866) | | | |
|---|---|---|---|---|---|---|---|---|---|---|---|---|
| | Cohen's d (95% CI) | p | p.adj | lnBF | Cohen's d (95% CI) | p | p.adj | lnBF | Cohen's d (95% CI) | p | p.adj | lnBF |
| CinguloOpercular-BrainStem | 0.203 (0.058, 0.347) | 0.006 | 0.167 | 1.204 | 0.096 (0.011, 0.181) | 0.028 | 0.144 | -0.611 | 0.125 (0.052, 0.199) | 0.001 | 0.013 | 2.436 |
| CinguloParietal-BrainStem | -0.258 (-0.404, -0.112) | 0.001 | 0.045 | 3.366 | -0.019 (-0.104, 0.067) | 0.672 | 0.878 | -2.924 | -0.032 (-0.106, 0.041) | 0.389 | 0.591 | -2.793 |
| CinguloParietal-RightPallidum | 0.222 (0.078, 0.366) | 0.003 | 0.086 | 1.974 | 0.112 (0.027, 0.197) | 0.010 | 0.096 | 0.262 | 0.097 (0.023, 0.17) | 0.010 | 0.062 | 0.151 |
| Default-DorsalAttention | -0.253 (-0.397, -0.109) | 0.001 | 0.045 | 3.290 | -0.071 (-0.156, 0.015) | 0.104 | 0.335 | -1.709 | -0.134 (-0.207, -0.061) | 0.000 | 0.007 | 3.284 |
| DorsalAttention-LeftHippocampus | -0.291 (-0.436, -0.146) | 0.000 | 0.027 | 5.060 | -0.037 (-0.123, 0.048) | 0.391 | 0.650 | -2.651 | -0.08 (-0.154, -0.007) | 0.032 | 0.124 | -0.884 |
| RetrosplenialTemporal-RightCerebellumCortex | 0.226 (0.082, 0.369) | 0.002 | 0.079 | 2.151 | 0.119 (0.033, 0.204) | 0.007 | 0.082 | 0.640 | 0.132 (0.058, 0.205) | 0.000 | 0.009 | 2.993 |
| Salience-LeftVentraldc | -0.242 (-0.386, -0.098) | 0.001 | 0.051 | 2.809 | -0.084 (-0.169, 0.001) | 0.053 | 0.222 | -1.159 | -0.121 (-0.195, -0.048) | 0.001 | 0.015 | 2.070 |
| SensorimotorHand-BrainStem | 0.247 (0.1, 0.393) | 0.001 | 0.051 | 2.854 | 0.096 (0.011, 0.181) | 0.027 | 0.144 | -0.595 | 0.122 (0.049, 0.195) | 0.001 | 0.015 | 2.136 |
| SensorimotorHand-RightHippocampus | 0.22 (0.074, 0.365) | 0.003 | 0.094 | 1.809 | -0.01 (-0.095, 0.075) | 0.820 | 0.940 | -2.992 | 0.038 (-0.035, 0.112) | 0.303 | 0.509 | -2.642 |
| Within CinguloOpercular | 0.229 (0.086, 0.372) | 0.002 | 0.074 | 2.323 | 0.129 (0.044, 0.214) | 0.003 | 0.048 | 1.350 | 0.144 (0.071, 0.217) | 0.000 | 0.004 | 4.254 |
| Within RetrosplenialTemporal | 0.266 (0.123, 0.409) | 0.000 | 0.042 | 3.989 | 0.034 (-0.051, 0.119) | 0.434 | 0.692 | -2.715 | 0.086 (0.013, 0.159) | 0.021 | 0.093 | -0.533 |

Student's and Bayes Factor T-tests were performed to compare controls with those with anhedonia. Student's t-statistic nominal p-values (p), conservative Benjamini-Hochberg adjusted p-values (p.adj) for 306 comparisons, and natural logarithms of Bayes Factors (lnBF) are reported. lnBF values greater than 1.1 were considered statistically significant. Note that the sum of the participants from the ABCD 1.0 and ABCD 4.0 (excluding) 1.0 samples exceeds the number of participants from the full ABCD 4.0 sample due to additional participants from the ABCD 1.0 sample being excluded from the full ABCD 4.0 sample during the quality control (QC) steps. Differences between processing pipelines between the data releases may account for these discrepancies in QC measures.

liberal adjusted p-value for each comparison. However, we decided to report the former since we did indeed perform 306 t-tests in our analyses. This reflects the somewhat arbitrary nature of statistical thresholding as the number of outcome measures considered in family-wise hypotheses can be difficult to clearly define [24].

To assess how well we reproduced the previous authors' results, we correlated our t-statistic values with those reported by the previous authors for the t-tests comparing individuals with or without anhedonia (S1A Fig—left). Similarly, we also correlated our lnBF statistics with those reported by the previous authors (S1A Fig—right). Of note, the previous authors also analyzed the associations between rsfMRI connectivity and depressed mood and anxiety in order to assess the specificity of their anhedonia results. We have also reproduced those same associations and have obtained correlations between our statistics and the previous authors' statistics for those analyses (S1B and S1C Fig). We note that although we used a more recent version of the R statistical software than the previous authors, the correlations between the statistics from our analyses and those from the previous authors were all equal to 1, indicating no differences in the results.

To check the assumptions for independent measures t-tests, we performed Shapiro-Wilk normality tests for rsfMRI measures from the ABCD 1.0 sample and found that all of the tests were significant, indicating non-normality (S1 Table). However, for large samples, such as in the ABCD study, statistical tests for normality are very sensitive to small deviations from normality which do not end up affecting the results of parametric tests. Furthermore, the central limit theorem states that for samples with n > 40, the means of random samples from any distribution tends to be normal, regardless of the distribution of the underlying data [25]. A visual inspection of the distributions, as well as the quantile-quantile (QQ) plots, for a representative sample of 3 of the rsfMRI measures significantly associated with anhedonia revealed no significant deviations from normality (S2 Fig).

We also performed F-tests to compare variances across the anhedonia and control groups for our t-tests in order to assess the assumptions of homoscedasticity. We note that there were 11 rsfMRI measures that exhibited significant F-tests in the ABCD 1.0 sample (S1 Table). To assess the potential bias introduced by unequal variances, we performed Welch's t-tests for those rsfMRI measures with significant F-tests and then correlated the Welch's t-statistics with Student's t-statistics. We then correlated the p-values from the two tests as well. We found significantly high correlations (r > = 0.99) for t-statistics and p-values, indicating minimal differences between the two types of tests and thus minimal impact of unequal variances on the results (S3A Fig).

For t-tests using the ABCD 1.0 sample, 288 participants with either missing rsfMRI data or missing youth-reported anhedonia responses were removed and 2437 were retained. Participants who were removed were on average 1 week younger than those who were retained (S10 Table). Such a difference was deemed to be negligible.

## Replication of previous findings

At the time of writing, the ABCD 4.0 release has been made available. We wanted to *replicate* the previous findings by using the full cohort, excluding the subjects used in the previous analyses, which is similar to replication in an independent sample. In this sub-sample of the ABCD 4.0 release (which excludes participants from ABCD 1.0 release), we found 591 participants who endorsed past and/or present anhedonia and 5,865 controls who did not. Demographic characteristics for this sample can be found in Table 3.

When using the ABCD 4.0 (excluding 1.0) sample, we found large reductions in effect sizes for all 11 rsfMRI measures previously found to be associated with anhedonia using the ABCD

**Table 3. Comparison of sociodemographic measures between controls and those with anhedonia in the ABCD 4.0 release, excluding ABCD 1.0 release, sub-sample.**

| Measures | Control (N = 5863) | Anhedonia (N = 591) | Statistic | p-value |
|---|---|---|---|---|
| % Male | 51.2 | 56.7 | 6.202 | 0.013 |
| Mean Age (months)[a] | 118.5 | 119.1 | -1.705 | 0.089 |
| % Asian | 2.3 | 0.8 | 4.703 | 0.03 |
| % Black | 15.2 | 20.8 | 12.449 | <0.001 |
| % Hispanic | 20.5 | 26.4 | 11.087 | 0.001 |
| % Other | 10.4 | 12.9 | 3.055 | 0.08 |
| % White | 51.6 | 39.1 | 33.287 | <0.001 |
| % Bipolar II | 0.3 | 8.3 | 313.34 | <0.001 |
| % Depressed Mood | 6.4 | 31.8 | 431.052 | <0.001 |
| % Irritability | 4.4 | 27.9 | 484.563 | <0.001 |

The proportion (%) of participants in each group for each measure are shown. Student's t-test was done to compare age (in months) between control and anhedonia groups. Chi-square tests of independence were done for all other measures. Note there is a slightly lower number of controls here than reported above due to the exclusion of participants with missing sociodemographic and/or comorbid psychiatric syndrome and diagnoses measures.

[a] Student's t-statistic.

1.0 sample (Table 2, center). Only one measure, the *Within-Cingulo-Opercular* connectivity measure, exhibited a statistically significant association as indicated by the adjusted p-value and lnBF statistics. However, as noted earlier, we applied fairly conservative corrections to the nominal p-values to adjust for multiple comparisons. At the nominal p-value level, 4 additional rsfMRI measures (the *Retrosplenial-Temporal vs. Right-Cerebellum-Cortex*, *Cingulo-Parietal vs. Right-Pallidum*, *Sensorimotor-Hand vs. BrainStem*, and *Cingulo-Opercular vs. BrainStem* connectivity measures) were also significantly associated with anhedonia.

For these t-tests using the ABCD 4.0 (excluding 1.0) sample, 124 participants with either missing rsfMRI or youth-reported anhedonia responses were excluded and 6456 participants were retained. Notably, those that were removed exhibited significantly higher proportions of individuals identifying as African Americans (27.4% vs 15.7%) and individuals endorsing depressed mood (12.9%. vs. 8.7%) (S10 Table). While significant patterns of missingness suggest Missingness At Random (MAR), the proportion of missing participants was ~2% and thus would not benefit from approaches such as multiple imputation (typically requiring missingness >5%) and are unlikely to lead to significant bias in the results [26].

To increase our power to detect genuine associations with smaller effect sizes, we next performed our analyses using the full ABCD 4.0 release sample, including all participants from the ABCD 1.0 release. Since we are including the participants used in the original analyses, our analyses using the full ABCD 4.0 sample would not be an independent replication of the previous results. Nevertheless, the results will help with the assessment of the stability of the effect sizes and associations. In the full ABCD 4.0 sample, there were 800 participants who endorsed past and/or present anhedonia and 8066 controls who did not. Demographic characteristics for this sample can be found in Table 4.

When using the full ABCD 4.0 sample, 6 of the original 11 rsfMRI measures exhibited statistically significant associations with anhedonia as indicated by the adjusted p-values and lnBF statistics, including the previously reproduced and replicated association for the *Within-Cingulo-Opercular* connectivity measure. Of note, associations for 3 out of the 4 rsfMRI measures (*Retrosplenial-Temporal vs. Right-Cerebellum-Cortex*, *Cingulo-Opercular vs. Brain-Stem*, *Sensorimotor-Hand vs. Brain-Stem* connectivity measures) replicated at the nominal p-value level using the ABCD 4.0 (excluding 1.0) sample were among the 6 significant rsfMRI

**Table 4. Comparison of sociodemographic measures between controls and those with anhedonia in the Full ABCD 4.0 release sample.**

| Measures | Control (N = 8064) | Anhedonia (N = 800) | Statistic | p-value |
|---|---|---|---|---|
| % Male[a] | 51.3 | 55.8 | 5.589 | 0.018 |
| Mean Age (months) | 119.1 | 119.3 | -0.845 | 0.398 |
| % Asian[a] | 2.2 | 0.9 | 5.363 | 0.021 |
| % Black[a] | 13.3 | 19.8 | 25.19 | <0.001 |
| % Hispanic[a] | 20.4 | 26.6 | 16.579 | <0.001 |
| % Other | 10.1 | 12.1 | 2.916 | 0.088 |
| % White[a] | 54 | 40.6 | 52.043 | <0.001 |
| % Bipolar II[a] | 0.5 | 7.9 | 347.62 | <0.001 |
| % Depressed Mood[a] | 6.9 | 30.9 | 507.29 | <0.001 |
| % Irritability[a] | 4.4 | 27.4 | 627.02 | <0.001 |

The proportion (%) of participants in each group for each measure are shown. Student's t-test was done to compare age (in weeks) between control and anhedonia groups. Chi-square tests of independence were done for all other measures. Note there is a slightly lower number of controls here than reported below due to the exclusion of participants with missing sociodemographic and/or comorbid psychiatric syndrome and diagnoses measures.

[a] Student's t-statistic.

measures found in the full ABCD sample. Effect sizes were intermediary between those found in the prior analyses but were much closer to the smaller effect sizes from the replication analyses using the ABCD 4.0 (excluding 1.0) sample.

For the t-tests using the full ABCD 4.0 sample, 139 participants with either missing rsfMRI measures or youth-reported anhedonia responses were removed and 8866 participants were retained. Similar to the other ABCD samples, there were higher proportions of individuals identifying as African American and those endorsing depressed mood in the participants who were removed. However, the proportion of participants removed was ~ 1.5% of the total sample and thus unlikely to contribute to significant bias to the analyses or receive benefit from multiple imputation.

To check the assumptions for independent samples t-tests performed using the ABCD 4.0 (excluding 1.0) and full ABCD 4.0 samples, we visually inspected the distributions of a representative sample of 3 rsfMRI measures significantly associated with anhedonia in each ABCD sample, which revealed no significant deviations from normality (S4 and S5 Figs). However, there were 17 rsfMRI measures in the ABCD 4.0 (excluding 1.0) sample, and 21 rsfMRI measures in the full ABCD 4.0 sample that exhibited significant F-tests, suggesting the presence of unequal variances. We performed Welch's t-tests for those rsfMRI measures with significant F-tests and then correlated the Welch's t-statistics with Student's t-statistics. We then correlated the p-values from the two tests as well. We found significantly high correlations ($r \geq 0.99$) for all the correlations, indicating minimal differences between the two types of tests and thus minimal impact of any unequal variances on the results (S3B and S3C Fig).

## Psychiatric co-morbidities and the specificity of findings

It is important to note that individuals reporting anhedonia may also report other symptoms or psychiatric diagnoses. Thus, it is important to evaluate the specificity of the associations between rsfMRI connectivity and anhedonia. In order to identify psychiatric conditions significantly comorbid in individuals reporting anhedonia, we performed tetrachoric correlations between anhedonia and 33 additional psychiatric diagnoses and symptoms collected at baseline (Fig 1).

Three psychiatric conditions exhibited correlation coefficients greater than or equal to 0.5 with anhedonia (S11 Table): irritability, depressed mood, and bipolar II disorder. Thus, these

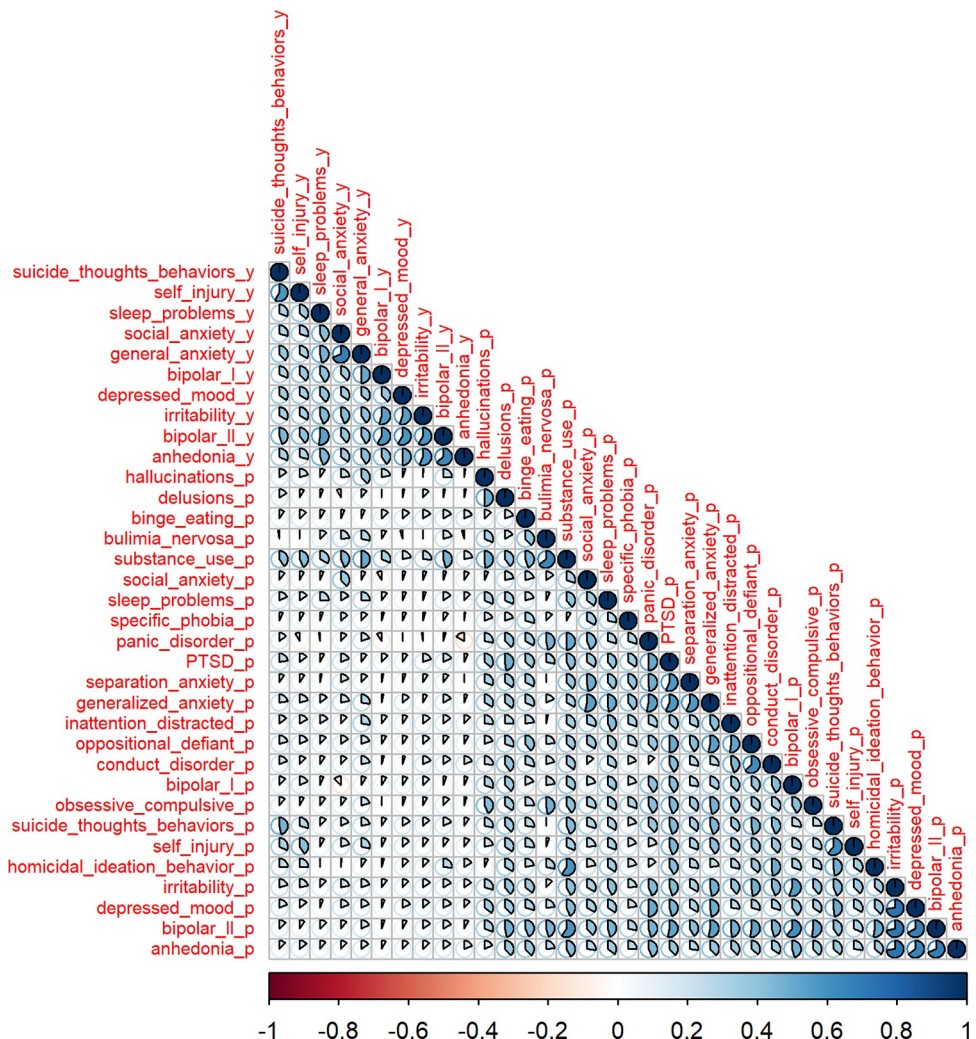

**Fig 1. Correlation matrix for 34 psychiatric symptoms and diagnoses based on youth and parent reports at baseline.** Tetrachoric correlation values are represented by the shaded proportion of the pie-charts in each cell of the correlation matrix as well as by the intensity of the shading on a divergent red and blue color-scale. Measures based on youths' self-reports end in the suffix "_y" while measures based on parent reports of their child end with the suffix "_p". Variables are ordered by hierarchical clustering.

were considered significantly co-morbid conditions that may potentially confound the associations between rsfMRI connectivity and anhedonia. In the full ABCD 4.0 sample, 32%, 28%, and 8% of the participants reporting anhedonia also reported depressed mood, irritability, and bipolar II disorder (S12 Table).

In line with the previous authors' approach, in order to assess the specificity of the associations found for anhedonia, we next performed t-tests to compare rsfMRI connectivity measures separately between those with and without symptoms of depressed mood, irritability, and bipolar II disorder using the full ABCD 4.0 sample. The presence of a psychiatric condition was the reference group.

We found that 2 rsfMRI measures significantly associated with anhedonia using the full ABCD 4.0 sample were also significantly associated with depressed mood ($n_{depressedMood}$ = 801, $n_{controls}$ = 8065). These were the *Default vs. Dorsal-Attention* (Cohen's d = -0.118, 95%CI [-0.191, -0.045], lnBF = 1.807) and *Within-Cingulo-Opercular* (Cohen's d = 0.117, 95%CI

[0.043, 0.19], lnBF = 1.646) connectivity networks (S13 Table). Similarly, 2 rsfMRI measures associated with anhedonia were also significantly associated with irritability ($n_{irritability}$ = 576, $n_{control}$ = 8290). These were the *Salience vs. Left-Ventraldc* (Cohen' d = -0.139, 95%CI [-0.224, -0.054], lnBF = 2.075) and *Default vs. Dorsal-Attention* (Cohen's d = -0.138, 95% CI [-0.223, -0.053], lnBF = 2.004) connectivity measures (S13 Table). None of the rsfMRI measures associated with anhedonia were also associated with bipolar II disorder ($n_{bipolarII}$ = 101, $n_{control}$ = 8765).

Altogether, the results of the t-test comparisons using the full ABCD 4.0 sample suggest the *Cingulo-Opercular vs. BrainStem*, *Retrosplenial-Temporal vs. Right-Cerebellum-Cortex, and the Sensorimotor-Hand vs. BrainStem* connectivity measures may be more specifically associated with anhedonia. Contrastingly, the *Within-Cingulo-Opercular* connectivity measure, whose association with anhedonia was the only one replicated at the more conservative adjusted p-value and lnBF statistic levels, was associated with both depressed mood and irritability.

## Multiple linear regression approach

While we were able to characterize which rsfMRI connectivity measures were more likely to be specifically associated with anhedonia compared to other psychiatric conditions, simple t-tests were not able to estimate the independent effects of each psychiatric condition on rsfMRI connectivity. For rsfMRI measures like the *Within-Cingulo-Opercular* connectivity measure, it would be important to disentangle its associations with anhedonia, depressed-mood, and irritability symptoms. Furthermore, t-tests are not able to control for potentially confounding sociodemographic variables such age, sex, and race/ethnicity. Indeed, Chi-square tests of independence showed significant differences in race/ethnicity and sex between individuals reporting anhedonia and those who do not across the different ABCD samples (Tables 1, 3, and 4).

By using a multiple linear regression approach where a rsfMRI connectivity measure is modeled as the response (or outcome) variable, comorbid psychiatric conditions as well as confounding factors can be included as explanatory (or predictor) variables. Thus, multiple linear regression allows for the estimation of the main effects of anhedonia on rsfMRI connectivity, independent of the effects of depressed mood, irritability, and bipolar II disorder (and vice versa), and the effects of confounding covariates. We hypothesized that controlling for potential sociodemographic confounders and accounting for the effects of co-morbid psychiatric conditions may improve replicability.

We took a step-wise approach and first only included the socio-demographic covariates (sex, age, race/ethnicity) in our regression models along with youth-reported anhedonia. Then, we added the comorbid psychiatric conditions (depressed mood, irritability, and bipolar II) to the models in order to evaluate their impact on the regression estimates and the replicability of any significant associations with anhedonia. We limited the inclusion of comorbid psychiatric conditions to these three measures in order to preserve the statistical power of our regression analyses and to account for comorbidities that are more likely to exhibit potential confounding effects based on their higher correlations with anhedonia (S11 Table).

Similar to the approach taken for replication analyses with t-statistics, we performed multiple linear regression in the ABCD 1.0 sample first to identify rsfMRI measures associated with anhedonia and then in the ABCD 4.0 (excluding 1.0) sample to replicate the results. For these analyses, the control group without symptoms of anhedonia was the reference group. We note that participants with missing covariate data were excluded from these analyses. Thus, the sample sizes will differ slightly from those used in the previous t-tests.

In the ABCD 1.0 release sample, there were 215 individuals who endorsed anhedonia and 2209 controls who did not. When we performed multiple linear regression with sex, age, and

race/ethnicity as covariates, 4 rsfMRI measures were significantly associated with anhedonia at the conservative adjusted p-value level and were among the 11 rsfMRI measures identified by the previous authors (Table 5, left). At the nominal p-value level, an additional 23 rsfMRI measures were found to be associated with anhedonia, including the remaining 7 rsfMRI measures identified by the previous authors. We note that the effect sizes were extremely small, with anhedonia accounting for less than 1% of the total variance in each rsfMRI measure.

We next attempted to replicate the associations using the ABCD 4.0 (excluding 1.0) sample where we identified 591 individuals with anhedonia and 5863 controls without. After performing the regressions, none of the 11 rsfMRI measures identified by the previous authors were significantly associated with anhedonia at either the adjusted or nominal p-value levels. However, 3 of the other rsfMRI measures that were significantly associated with anhedonia at the nominal p-value level using the ABCD 1.0 sample, were also found to be associated with anhedonia in the ABCD 4.0 (excluding 1.0) sample at the conservative adjusted p-value level (Table 5, center). The 3 replicated rsfMRI measures were the *Auditory vs. Right Putamen*, *Retrosplenial-Temporal vs. Right-Thalamus-Proper*, and *Salience vs. Right-Amygdala* connectivity measures.

Interestingly, while the effect sizes for the majority of rsfMRI measures reduced significantly after replication using the ABCD 4.0 (excluding 1.0) sample (most notably for rsfMRI measures with the largest effect sizes in the ABCD 1.0 sample analyses), the effect sizes for the 3 replicated rsfMRI measures remained roughly the same. However, the sign of the anhedonia partial regression coefficient for the *Salience vs. Right-Amygdala* measure flipped from being negative to positive. Since the regression model independent variables were not found to be significantly multicollinear (S14 and S15 Tables), it is more likely that this flip was driven by sample specific characteristics or random measurement error.

Finally, we performed regressions using the full ABCD 4.0 sample in order to increase our statistical power and to assess the stability of the effect sizes and associations. Note that the results cannot be considered an independent replication of the initial associations since we are including the participants from the ABCD 1.0 sample. When performing regressions in the full ABCD 4.0 sample, 11 of the 27 rsfMRI measures found to be associated with anhedonia in the ABCD 1.0 sample, were also associated with anhedonia at the adjusted p-value level, including the 3 rsfMRI measures replicated in the ABCD 4.0 (excluding 1.0) sample (Table 5, right). We note that the effect sizes were intermediary between those estimated from analyses using the ABCD 1.0 and ABCD 4.0 (excluding 1.0) samples, although most were closer to those from the latter.

Next, we added depressed-mood, irritability, and bipolar II disorder measures to the regression analyses across the ABCD samples in order to assess the impact of accounting for comorbid psychiatric conditions on the specificity and stability of the effect sizes and significance of the associations between rsfMRI measures and anhedonia that were previously identified.

For the *Auditory vs. Right-Putamen* and *Retrosplenial-Temporal vs. Right-Thalamus-Proper* measures, we found that their significant associations with anhedonia were preserved in the regressions using the ABCD 4.0 (excluding 1.0) and full ABCD 4.0 samples, even after accounting for psychiatric comorbidities (Table 6, center and right). Furthermore, their effect sizes remained relatively consistent across the analyses using the different ABCD samples. Importantly, depressed-mood, irritability, and bipolar II disorder were not found to be significant predictors of these two rsfMRI measures in any of the multiple linear regression analyses, suggesting these associations are specific to anhedonia (S17 Table).

We note that the *Auditory vs. Right Putamen* and *Retrosplenial-Temporal vs. Right-Thalamus-Proper* measures were not found to be associated with anhedonia in the ABCD 1.0 sample (Table 6, left) at the adjusted or nominal p-value levels (although there was a trend). However,

**Table 5. Effects of anhedonia from multiple linear regressions controlling for sociodemographic factors only across the ABCD study samples.**

| rsfMRI Connectivity | ABCD 1.0 Sample (n = 2424) | | | | | ABCD 4.0 (excluding 1.0) Sample (n = 6454) | | | | | Full ABCD 4.0 Sample (n = 8864) | | | | |
|---|---|---|---|---|---|---|---|---|---|---|---|---|---|---|---|
| | Effect Size (% variance) (95%CI) | Estimate | Std. Err | p | p.adj | Effect size (% variance) (95%CI) | Estimate | Std. Err | p | p.adj | Effect size (% variance) (95% CI) | Estimate | Std. Err | p | p.adj |
| Auditory-LeftAmygdala | 0.21 (0.01, 0.71) | 0.146 | 0.074 | 0.049 | 0.228 | 0.02 (0, 0.12) | 0.038 | 0.044 | 0.385 | 0.527 | 0.04 (0, 0.18) | 0.066 | 0.038 | 0.080 | 0.147 |
| Auditory-RightPutamen | 0.22 (0.01, 0.79) | -0.143 | 0.073 | 0.050 | 0.228 | **0.21 (0.04, 0.5)** | **-0.122** | **0.043** | **0.004** | **0.013** | **0.19 (0.06, 0.44)** | **-0.120** | **0.037** | **0.001** | **0.003** |
| CinguloOpercular-BrainStem | 0.35 (0.03, 1.02) | -0.191 | 0.073 | 0.009 | 0.076 | 0.06 (0, 0.22) | -0.060 | 0.043 | 0.163 | 0.275 | **0.11 (0.02, 0.29)** | **-0.092** | **0.037** | **0.012** | **0.029** |
| CinguloOpercular-LeftAccumbensArea | 0.24 (0.01, 0.79) | -0.160 | 0.074 | 0.031 | 0.169 | 0.03 (0, 0.16) | -0.047 | 0.043 | 0.275 | 0.413 | **0.1 (0.01, 0.26)** | **-0.095** | **0.037** | **0.010** | **0.024** |
| CinguloOpercular-RightVentraldc | 0.27 (0.02, 0.85) | -0.166 | 0.075 | 0.026 | 0.157 | 0.02 (0, 0.16) | -0.027 | 0.043 | 0.524 | 0.659 | 0.05 (0.01, 0.19) | -0.055 | 0.037 | 0.134 | 0.225 |
| CinguloParietal-BrainStem | **0.41 (0.05, 1.13)** | **0.219** | **0.075** | **0.003** | **0.037** | 0 (0, 0.09) | 0.003 | 0.044 | 0.946 | 0.972 | 0.01 (0, 0.09) | 0.019 | 0.038 | 0.614 | 0.729 |
| CinguloParietal-RightPallidum | 0.36 (0.03, 1.01) | -0.193 | 0.074 | 0.009 | 0.074 | 0.08 (0.01, 0.27) | -0.071 | 0.043 | 0.096 | 0.180 | 0.06 (0.01, 0.21) | -0.054 | 0.037 | 0.141 | 0.234 |
| Default-DorsalAttention | **0.44 (0.07, 1.16)** | **0.217** | **0.073** | **0.003** | **0.034** | 0.03 (0, 0.17) | 0.041 | 0.043 | 0.342 | 0.486 | **0.11 (0.02, 0.29)** | **0.096** | **0.037** | **0.009** | **0.021** |
| Default-LeftHippocampus | 0.28 (0.01, 0.85) | -0.173 | 0.075 | 0.021 | 0.136 | 0.02 (0, 0.14) | -0.030 | 0.043 | 0.480 | 0.618 | 0.06 (0, 0.2) | -0.068 | 0.037 | 0.067 | 0.126 |
| Default-LeftVentraldc | 0.19 (0, 0.67) | -0.152 | 0.075 | 0.043 | 0.208 | 0.03 (0, 0.17) | -0.042 | 0.044 | 0.331 | 0.473 | 0.06 (0, 0.22) | -0.074 | 0.037 | 0.048 | 0.093 |
| DorsalAttention-LeftAccumbensArea | 0.25 (0.02, 0.77) | -0.161 | 0.074 | 0.029 | 0.166 | 0 (0, 0.09) | 0.019 | 0.043 | 0.658 | 0.773 | 0.01 (0, 0.11) | -0.019 | 0.037 | 0.603 | 0.721 |
| DorsalAttention-LeftHippocampus | **0.54 (0.1, 1.36)** | **0.251** | **0.074** | **0.001** | **0.013** | 0.01 (0, 0.1) | 0.004 | 0.043 | 0.921 | 0.956 | 0.04 (0, 0.18) | 0.049 | 0.037 | 0.188 | 0.294 |
| DorsalAttention-LeftVentraldc | 0.24 (0.01, 0.81) | -0.155 | 0.074 | 0.036 | 0.186 | 0.07 (0, 0.27) | -0.082 | 0.044 | 0.058 | 0.120 | **0.1 (0.01, 0.27)** | **-0.097** | **0.037** | **0.009** | **0.022** |
| FrontoParietalLeftVentraldc | 0.21 (0.01, 0.73) | -0.157 | 0.076 | 0.039 | 0.195 | 0 (0, 0.09) | 0.010 | 0.044 | 0.827 | 0.893 | 0.01 (0, 0.09) | -0.014 | 0.038 | 0.715 | 0.809 |
| FrontoParietal-RightVentraldc | 0.3 (0.01, 1.04) | -0.192 | 0.076 | 0.012 | 0.090 | 0 (0, 0.09) | 0.002 | 0.044 | 0.968 | 0.983 | 0.01 (0, 0.1) | -0.033 | 0.038 | 0.378 | 0.511 |
| FrontoParietal-SensorimotorMouth | 0.32 (0.02, 0.95) | -0.195 | 0.074 | 0.009 | 0.074 | 0.03 (0, 0.17) | -0.054 | 0.043 | 0.212 | 0.338 | 0.02 (0, 0.12) | -0.037 | 0.037 | 0.326 | 0.460 |
| RetrosplenialTemporal-RightCerebellumCortex | 0.34 (0.04, 0.91) | -0.180 | 0.073 | 0.014 | 0.100 | 0.09 (0.01, 0.29) | -0.072 | 0.043 | 0.091 | 0.174 | **0.13 (0.03, 0.31)** | **-0.093** | **0.037** | **0.012** | **0.027** |
| RetrosplenialTemporal-RightThalamusProper | 0.26 (0.01, 0.93) | -0.162 | 0.073 | 0.027 | 0.160 | **0.21 (0.04, 0.52)** | **-0.133** | **0.043** | **0.002** | **0.006** | **0.25 (0.08, 0.54)** | **-0.152** | **0.037** | **0.000** | **0.000** |
| Salience-LeftAccumbensArea | 0.22 (0.01, 0.7) | -0.153 | 0.074 | 0.038 | 0.192 | 0.05 (0, 0.22) | -0.057 | 0.043 | 0.188 | 0.306 | 0.07 (0.01, 0.22) | -0.067 | 0.037 | 0.071 | 0.131 |

*(Continued)*

**Table 5.** (Continued)

| rsfMRI Connectivity | ABCD 1.0 Sample (n = 2424) | | | | | ABCD 4.0 (excluding 1.0) Sample (n = 6454) | | | | | Full ABCD 4.0 Sample (n = 8864) | | | | |
|---|---|---|---|---|---|---|---|---|---|---|---|---|---|---|---|
| | Effect Size (% variance) (95%CI) | Estimate | Std. Err | p | p.adj | Effect size (% variance) (95%CI) | Estimate | Std. Err | p | p.adj | Effect size (% variance) (95% CI) | Estimate | Std. Err | p | p.adj |
| Salience-LeftVentraldc | 0.33 (0.02, 0.97) | 0.188 | 0.074 | 0.011 | 0.087 | 0.05 (0, 0.23) | 0.060 | 0.043 | 0.165 | 0.278 | **0.1 (0.02, 0.3)** | **0.091** | **0.037** | **0.015** | **0.033** |
| Salience-RightAmygdala | 0.18 (0, 0.73) | -0.146 | 0.074 | 0.047 | 0.225 | **0.12 (0.01, 0.38)** | **0.126** | **0.044** | **0.004** | **0.011** | **0.08 (0.01, 0.24)** | **0.105** | **0.037** | **0.005** | **0.012** |
| SensorimotorHand-BrainStem | 0.32 (0.04, 0.87) | -0.186 | 0.074 | 0.012 | 0.094 | 0.07 (0, 0.25) | -0.064 | 0.043 | 0.133 | 0.235 | **0.11 (0.02, 0.3)** | **-0.088** | **0.037** | **0.017** | **0.038** |
| SensorimotorHand-RightHippocampus | 0.32 (0.02, 0.89) | -0.179 | 0.074 | 0.016 | 0.114 | 0.01 (0, 0.14) | 0.047 | 0.043 | 0.273 | 0.413 | 0.01 (0, 0.09) | -0.006 | 0.037 | 0.874 | 0.924 |
| SensorimotorHand-RightThalamusProper | 0.2 (0, 0.74) | 0.162 | 0.075 | 0.030 | 0.167 | 0 (0, 0.09) | -0.008 | 0.043 | 0.855 | 0.913 | 0.03 (0, 0.15) | 0.054 | 0.037 | 0.146 | 0.242 |
| SensorimotorMouth-LeftVentraldc | 0.25 (0.01, 0.87) | -0.158 | 0.073 | 0.031 | 0.171 | 0.01 (0, 0.09) | 0.005 | 0.042 | 0.905 | 0.946 | 0.05 (0, 0.18) | -0.054 | 0.037 | 0.141 | 0.234 |
| Within-CinguloOpercular | 0.39 (0.05, 1.06) | -0.183 | 0.072 | 0.011 | 0.088 | 0.1 (0.01, 0.31) | -0.081 | 0.043 | 0.057 | 0.117 | **0.13 (0.03, 0.32)** | **-0.093** | **0.036** | **0.011** | **0.026** |
| Within-RetrosplenialTemporal | **0.57 (0.17, 1.34)** | **-0.255** | **0.073** | **0.000** | **0.009** | 0.01 (0, 0.09) | 0.007 | 0.043 | 0.863 | 0.919 | 0.05 (0, 0.17) | -0.048 | 0.037 | 0.188 | 0.294 |

Effect size represents the percentage of the total variance (proportion of variance * 100%) in a rsfMRI connectivity measure accounted for by anhedonia. The partial regression coefficient (Estimate), standard error (Std.Err), nominal p-value (p), and conservative Benjamini-Hochberg adjusted p-value (p.adj) for anhedonia are also shown for each rsfMRI regression model. Partial regression coefficient estimates represent changes in the rsfMRI measure, in standard deviation units from the mean, associated with the presence of anhedonia. Bolded results were significant at the adjusted p-value level (p.adj < 0.05). Note that the sum of the participants from the ABCD 1.0 and ABCD 4.0 (excluding) 1.0 samples exceeds the number of participants from the full ABCD 4.0 sample due to additional participants from the ABCD 1.0 sample being excluded from the full ABCD 4.0 sample during the quality control (QC) steps. Differences between processing pipelines between the data releases may account for these discrepancies in QC measures.

we see that the effect sizes and partial regression coefficients were similar in magnitude and sign to those estimated in the corresponding regressions using the ABCD 4.0 (excluding 1.0) and full ABCD 4.0 samples, albeit with wider confidence intervals and larger standard errors. These patterns suggest that the ABCD 1.0 sample size was not well-powered enough, after including the additional psychiatric comorbidity measures, to detect these associations.

Anhedonia was also found to be significantly associated with the *CinguloOpercular-Brainstem* connectivity measure at the adjusted p-value level when using the full ABCD 4.0 sample (Table 6, right). However, as with most of the other rsfMRI measures, its effect size was significantly decreased in the replication analyses using the ABCD 4.0 (excluding 1.0) sample. The instability in effect size suggests that the association found using the full ABCD 4.0 sample is likely driven by the inflated effect size found in the ABCD 1.0 sample which, in turn, may be due to sample-specific characteristics (ie. other confounding factors) or random measurement error.

The remaining rsfMRI measures exhibited extremely small effect sizes that did not survive statistical thresholding. However, larger samples with more statistical power may be able to

**Table 6. Effects of anhedonia from multiple linear regressions controlling for sociodemographic factors and clinical comorbidities across the ABCD study samples.**

| rsfMRI connectivity | ABCD 1.0 Sample (n = 2424) | | | | | ABCD 4.0 (excluding 1.0) Sample (n = 6454) | | | | | Full ABCD 4.0 Sample (n = 8864) | | | | |
|---|---|---|---|---|---|---|---|---|---|---|---|---|---|---|---|
| | Effect size (% variance) (95%CI) | Estimate | Std. Err | P | p.adj | Effect size (% variance) (95%CI) | Estimate | Std. Err | p | p.adj | Effect size (% variance) (95%CI) | Estimate | Std. Err | p-value | p.adj |
| Auditory-LeftAmygdala | 0.19 (0.01, 0.65) | 0.144 | 0.078 | 0.066 | 0.339 | 0.01 (0, 0.13) | 0.031 | 0.047 | 0.514 | 0.723 | 0.03 (0, 0.16) | 0.052 | 0.040 | 0.189 | 0.365 |
| Auditory-RightPutamen | 0.21 (0.01, 0.79) | -0.144 | 0.076 | 0.059 | 0.325 | **0.18 (0.04, 0.49)** | **-0.118** | **0.046** | **0.010** | **0.037** | **0.18 (0.05, 0.38)** | **-0.119** | **0.039** | **0.002** | **0.009** |
| CinguloOpercular-BrainStem | 0.37 (0.04, 1.02) | -0.211 | 0.077 | 0.006 | 0.075 | 0.06 (0, 0.21) | -0.070 | 0.046 | 0.125 | 0.283 | **0.1 (0.02, 0.25)** | **-0.095** | **0.039** | **0.015** | **0.047** |
| CinguloOpercular-LeftAccumbensArea | 0.19 (0.01, 0.72) | -0.127 | 0.078 | 0.104 | 0.436 | 0.02 (0, 0.14) | -0.038 | 0.046 | 0.407 | 0.628 | 0.08 (0.01, 0.22) | -0.084 | 0.039 | 0.034 | 0.094 |
| CinguloOpercular-RightVentraldc | 0.23 (0.02, 0.77) | -0.142 | 0.078 | 0.070 | 0.352 | 0.02 (0, 0.13) | -0.023 | 0.046 | 0.620 | 0.798 | 0.04 (0.01, 0.15) | -0.040 | 0.039 | 0.308 | 0.510 |
| CinguloParietal-BrainStem | **0.41 (0.05, 1.16)** | **0.232** | **0.078** | **0.003** | **0.049** | 0 (0, 0.1) | -0.008 | 0.047 | 0.865 | 0.939 | 0.01 (0, 0.09) | 0.023 | 0.040 | 0.561 | 0.746 |
| CinguloParietal-RightPallidum | 0.28 (0.03, 0.9) | -0.154 | 0.077 | 0.046 | 0.283 | 0.07 (0.01, 0.24) | -0.068 | 0.046 | 0.136 | 0.302 | 0.05 (0.01, 0.18) | -0.049 | 0.039 | 0.207 | 0.390 |
| Default-DorsalAttention | 0.4 (0.06, 1.06) | 0.204 | 0.076 | 0.007 | 0.086 | 0.01 (0, 0.12) | -0.001 | 0.046 | 0.978 | 0.993 | 0.07 (0.01, 0.22) | 0.054 | 0.039 | 0.166 | 0.334 |
| Default-LeftHippocampus | 0.3 (0.03, 0.93) | -0.195 | 0.078 | 0.013 | 0.129 | 0.01 (0, 0.12) | -0.020 | 0.046 | 0.661 | 0.821 | 0.05 (0, 0.17) | -0.061 | 0.039 | 0.117 | 0.258 |
| Default-LeftVentraldc | 0.17 (0.01, 0.6) | -0.136 | 0.078 | 0.083 | 0.389 | 0.03 (0, 0.18) | -0.059 | 0.047 | 0.209 | 0.410 | 0.06 (0, 0.21) | -0.074 | 0.040 | 0.063 | 0.157 |
| DorsalAttention-LeftAccumbensArea | 0.31 (0.02, 0.91) | -0.210 | 0.077 | 0.007 | 0.083 | 0 (0, 0.08) | 0.021 | 0.046 | 0.651 | 0.818 | 0.01 (0, 0.09) | -0.012 | 0.039 | 0.765 | 0.879 |
| DorsalAttention-LeftHippocampus | **0.51 (0.07, 1.23)** | **0.250** | **0.078** | **0.001** | **0.028** | 0.01 (0, 0.08) | -0.033 | 0.046 | 0.476 | 0.689 | 0.03 (0, 0.13) | 0.022 | 0.039 | 0.573 | 0.757 |
| DorsalAttention-LeftVentraldc | 0.25 (0.02, 0.87) | -0.171 | 0.077 | 0.027 | 0.210 | 0.06 (0, 0.25) | -0.067 | 0.047 | 0.149 | 0.321 | 0.08 (0.01, 0.26) | -0.082 | 0.040 | 0.038 | 0.105 |
| FrontoParietal-LeftVentraldc | 0.22 (0.01, 0.78) | -0.168 | 0.079 | 0.034 | 0.236 | 0 (0, 0.08) | 0.004 | 0.047 | 0.941 | 0.973 | 0 (0, 0.08) | -0.011 | 0.040 | 0.783 | 0.888 |
| FrontoParietal-RightVentraldc | 0.32 (0.02, 1) | -0.210 | 0.080 | 0.008 | 0.095 | 0 (0, 0.08) | 0.004 | 0.047 | 0.937 | 0.972 | 0.01 (0, 0.1) | -0.034 | 0.040 | 0.398 | 0.601 |
| FrontoParietal-SensorimotorMouth | 0.29 (0.02, 0.81) | -0.189 | 0.078 | 0.015 | 0.144 | 0.03 (0, 0.17) | -0.067 | 0.047 | 0.150 | 0.323 | 0.03 (0, 0.13) | -0.056 | 0.040 | 0.160 | 0.326 |
| RetrosplenialTemporal-RightCerebellumCortex | 0.28 (0.04, 0.85) | -0.154 | 0.077 | 0.044 | 0.277 | 0.08 (0.01, 0.28) | -0.075 | 0.046 | 0.104 | 0.247 | 0.12 (0.03, 0.28) | -0.094 | 0.039 | 0.016 | 0.050 |
| RetrosplenialTemporal-RightThalamusProper | 0.21 (0.01, 0.75) | -0.135 | 0.077 | 0.079 | 0.377 | **0.17 (0.03, 0.44)** | **-0.122** | **0.046** | **0.008** | **0.029** | **0.21 (0.07, 0.44)** | **-0.140** | **0.039** | **0.000** | **0.002** |
| Salience-LeftAccumbensArea | 0.19 (0.01, 0.69) | -0.134 | 0.077 | 0.082 | 0.386 | 0.04 (0, 0.18) | -0.039 | 0.046 | 0.403 | 0.624 | 0.06 (0.01, 0.2) | -0.055 | 0.039 | 0.163 | 0.330 |
| Salience-LeftVentraldc | 0.33 (0.03, 1) | 0.199 | 0.077 | 0.010 | 0.113 | 0.03 (0, 0.18) | 0.032 | 0.046 | 0.493 | 0.703 | 0.07 (0.01, 0.23) | 0.062 | 0.039 | 0.114 | 0.251 |
| Salience-RightAmygdala | 0.21 (0.01, 0.77) | -0.173 | 0.077 | 0.026 | 0.203 | 0.1 (0, 0.3) | 0.111 | 0.047 | 0.017 | 0.058 | 0.07 (0, 0.23) | 0.095 | 0.040 | 0.017 | 0.052 |
| SensorimotorHand-BrainStem | 0.25 (0.02, 0.78) | -0.147 | 0.077 | 0.058 | 0.321 | 0.05 (0.01, 0.21) | -0.048 | 0.046 | 0.298 | 0.518 | 0.08 (0.01, 0.23) | -0.068 | 0.039 | 0.082 | 0.195 |
| SensorimotorHand-RightHippocampus | 0.29 (0.03, 0.88) | -0.174 | 0.078 | 0.026 | 0.204 | 0.01 (0, 0.12) | 0.044 | 0.046 | 0.339 | 0.561 | 0.01 (0, 0.08) | 0.002 | 0.039 | 0.957 | 0.981 |
| SensorimotorHand-RightThalamusProper | 0.19 (0.01, 0.67) | 0.157 | 0.078 | 0.045 | 0.280 | 0.01 (0, 0.11) | -0.045 | 0.047 | 0.337 | 0.561 | 0.02 (0, 0.14) | 0.024 | 0.040 | 0.538 | 0.728 |
| SensorimotorMouth-LeftVentraldc | 0.22 (0.01, 0.8) | -0.140 | 0.076 | 0.068 | 0.347 | 0.01 (0, 0.09) | -0.016 | 0.046 | 0.726 | 0.864 | 0.04 (0, 0.15) | -0.060 | 0.039 | 0.126 | 0.273 |

*(Continued)*

**Table 6.** (Continued)

| rsfMRI connectivity | ABCD 1.0 Sample (n = 2424) | | | | | ABCD 4.0 (excluding 1.0) Sample (n = 6454) | | | | | Full ABCD 4.0 Sample (n = 8864) | | | | |
|---|---|---|---|---|---|---|---|---|---|---|---|---|---|---|---|
| | Effect size (% variance) (95%CI) | Estimate | Std. Err | P | p.adj | Effect size (% variance) (95%CI) | Estimate | Std. Err | p | p.adj | Effect size (% variance) (95%CI) | Estimate | Std. Err | p-value | p.adj |
| Within-CinguloOpercular | 0.3 (0.03, 0.86) | -0.141 | 0.076 | 0.062 | 0.332 | 0.08 (0.01, 0.25) | -0.056 | 0.046 | 0.217 | 0.420 | 0.1 (0.02, 0.26) | -0.068 | 0.039 | 0.078 | 0.188 |
| Within-RetrosplenialTemporal | 0.44 (0.08, 1.07) | -0.204 | 0.076 | 0.008 | 0.088 | 0 (0, 0.08) | 0.038 | 0.046 | 0.406 | 0.627 | 0.03 (0.01, 0.14) | -0.014 | 0.039 | 0.728 | 0.857 |

Effect size represents the percentage of the total variance (proportion of variance * 100%) in a rsfMRI connectivity measure accounted for by anhedonia. The partial regression coefficient (Estimate), standard error (Std.Err), nominal p-value (p), and conservative Benjamini-Hochberg adjusted p-value (p.adj) for anhedonia are also shown for each rsfMRI regression model. Partial regression coefficient estimates represent changes in the rsfMRI measure, in standard deviation units from the mean, associated with the presence of anhedonia. Bolded results were significant at the adjusted p-value level (p.adj < 0.05). Note that the sum of the participants from the ABCD 1.0 and ABCD 4.0 (excluding) 1.0 samples exceeds the number of participants from the full ABCD 4.0 sample due to additional participants from the ABCD 1.0 sample being excluded from the full ABCD 4.0 sample during the quality control (QC) steps. Differences between processing pipelines between the data releases may account for these discrepancies in QC measures.

detect them, though whether these effects, along with the ones we detected in this study, are clinically meaningful remains to be determined.

We note that there were 2163 families in the ABCD 1.0 sample, 5754 families in the ABCD 4.0 (excluding 1.0) sample, and 7688 families in the ABCD 4.0 sample. The average Intra-Class-Correlation (ICC) across the regressions using the ABCD 1.0 sample was 0.09 (sd = 0.07) indicating that the random effects of *family* structure accounted for about 9% of the variability in rsfMRI connectivity measures (S4 and S5 Tables). However, the standard deviation was fairly large, indicating significant variability across regressions. Similarly, for regressions using the ABCD 4.0 (excluding 1.0) sample and the full ABCD 4.0 sample, the average ICC across regressions was 0.10 (SD = 0.05) and 0.11 (SD = 0.05), respectively (S6–S9 Tables).

For the multiple linear regression analyses, patterns of missingness closely resembled those reported for the samples used in the t-tests. Notably, individuals removed due to missingness exhibited higher proportions of those identifying as African American and those endorsing depressed mood (S16 Table). However, the proportion of participants removed was less than 5% of the total sample for each set of analyses and thus, were unlikely to bias the results.

Several assumptions for multiple linear regression were assessed across all the analyses. Multicollinearity between predictors was assessed with the variance inflation factor (VIF) where generalized VIF (GVIF) values greater than 5 indicate significant multicollinearity between predictors in a multiple linear regression model. For the regressions performed across the ABCD 1.0, ABCD 4.0 (excluding 1.0), and full ABCD 4.0 samples, we obtained GVIF values for predictors in each regression and then found their average values across all regressions, separately for each ABCD sample (S14 and S15 Tables). No regressions exhibited significant multicollinearity between predictors. We performed Durbin-Watson tests to detect auto-correlation between the residuals from each of the regressions and found all test statistics were between 1.5–2.5, and thus within the acceptable range for auto-correlations (S4–S9 Tables).

We performed the Breusch-Pagan (BP) test to the assumption of homoscedasticity for the model residuals from all the regressions (S4–S9 Tables). For the regressions using the ABCD 1.0 sample, one rsfMRI measure associated with anhedonia exhibited a significant BP test. For the regressions using the ABCD 4.0 (excluding 1.0) and full ABCD 4.0 samples, 4 rsfMRI

measures associated with anhedonia exhibited significant BP tests, each. For the rsfMRI measures with significant BP tests, we plotted their model residuals against marginal model fitted values and did not detect any significant patterns of heteroskedasticity, visually (S6–S11 Figs (left)). To be sure, we performed weighted-least-squares (WLS) regressions, which is able to account for differences in variance in the residuals, for these rsfMRI measures, and then correlated the t-statistics from the WLS and the original ordinary-least squares (OLS) regressions. The correlations were all equal to 1, indicating little effect of potential unequal variances on the results (S6–S11 Figs (right)).

Finally, we performed a visual inspection of density plots of the residuals from regressions for 3 representative rsfMRI measures significantly associated with anhedonia and their Quantile-Quantile (QQ) plots for each set of regressions across the ABCD samples. All appeared normally distributed (S12–S17 Figs.).

## Discussion

### Reproduction and replication of previous findings

While we were able to successfully reproduce the previous authors' findings, we were mostly unable to replicate them using a larger independent subset of the full ABCD 4.0 release sample. Using t-tests, only the *Within-Cingulo-Opercular* rsfMRI measure was consistently associated with anhedonia across the ABCD 4.0 (excluding 1.0) and full ABCD 4.0 samples. However, when we controlled for demographic covariates (sex, age, and race/ethnicity) using a linear regression approach, the association was no longer replicable in the ABCD 4.0 (excluding 1.0) sample. Like the other associations identified by the previous authors, we observed a significant decrease in effect size in the replication analysis after controlling for these additional covariates suggesting the presence of significant confounding effects that were not accounted for with t-tests. Furthermore, large decreases in effect size in replication analyses have been reported to occur more frequently when the initial discovery sample is small, such that the most inflated and statistically significant findings are the most likely to be identified and reported [27]. Unfortunately, these inflated findings are the least replicable as regression towards the mean leads to reductions in both effect size and significance in subsequent replications. Of note, Marek et al., 2022 found that controlling for sociodemographic covariates generally reduced effect sizes, and thus may help reduce effect size inflation.

Interestingly, when we considered rsfMRI measures associated with anhedonia at the nominal p-value level in our multiple linear regressions using the ABCD 1.0 sample, we were able to identify 2 new rsfMRI measures (the *Auditory vs. Right Putamen* and the *Retrosplenial-Temporal vs. Right-Thalamus-Proper* measures) associated with anhedonia that were replicable. Notably, these 2 rsfMRI measures exhibited smaller, and less statistically significant, effect sizes in the ABCD 1.0 sample that remained relatively consistent across the regressions using the ABCD 4.0 (excluding 1.0) and full ABCD 4.0 samples. Thus, these findings suggest that replicability may be improved if less emphasis was placed on associations with the most statistical significance but rather on identifying those with the most stable effects across analyses, even if they are less statistically significant.

Of note, a recent study exploring the replicability of brain-behavior association studies using simulations and parametric bootstrapping methods found that relatively small sample sizes (n<500) produced results with significantly inflated effect sizes, low precision, and low replicability and it was only when the sample sizes were increased to the high hundreds or thousands were they able to produce stable effects that were significantly more replicable [27]. To maximize statistical power and to reduce the likelihood of selecting inflated associations, future discovery analyses should be conducted using larger samples, such as in the full ABCD 4.0 release sample.

Although we found 2 rsfMRI measures with replicable associations, the actual effect sizes were extremely small, with anhedonia accounting for about 0.2% of the total variance for each rsfMRI measure. By themselves, these findings are unlikely to be clinically meaningful. However, combining the small effects of many brain-based measures together may produce signals with significant clinical utility in either diagnostic prediction or monitoring disease progression [28, 29]. An analogous approach has been taken in genetic research whereby many genetic variants, which individually exhibit miniscule amounts of association, can be combined to produce genetic risk scores [30] that altogether contribute meaningfully to the prediction of the course of complex neuropsychiatric disorders [31].

By including the other predictors, the multiple linear regression models accounted for about 4.5% of the total variance in the *Auditory vs. Right Putamen* rsfMRI measure and about 3% of the total variance for the *Retrosplenial-Temporal vs. Right-Thalmaus-Proper* rsfMRI measure (Fig 2). For both regression models, the race/ethnicity predictor accounted for the vast majority of the explained variance. Upon inspection of the partial regression coefficients for each model, Black, Hispanic, and Other race/ethnicity exhibited the largest and most significant partial regression coefficients for both rsfMRI measures (S9 Table).

Importantly, we also found there were significantly higher proportions of Black and Hispanic participants in the anhedonia group compared to controls without anhedonia in the ABCD samples. Race and ethnicity are social constructs representing complex social and cultural factors [32] deserving careful consideration. Several previous studies have reported significantly higher risk of anhedonia in Black and Hispanic compared to non-Hispanic White adults [33, 34] and that these associations may, in part, be accounted for by socioeconomic factors, such as household income and education, as well as other social determinants of health [35], such as disparities in access to healthcare [33]. In the ABCD sample, racial discrimination may be an important factor contributing to risk of anhedonia as well as differences in brain-based measures. For example, several recent studies have found that racial discrimination is associated with lower total brain volume [36] and alterations in prefrontal white matter tracts in adults [37, 38]. While out of the scope of this study, it will be critical to investigate how social determinants of health and other environmental factors, such as trauma [4, 39], contribute to differences in health and brain-based outcomes between different racial and ethnic groups during child and adolescent development.

## Specificity of associations

We found depressed mood, irritability, and bipolar II disorder to be significantly comorbid with anhedonia. Using a multiple linear regression approach in the full ABCD 4.0 dataset, we were able to estimate the effects of anhedonia on rsfMRI connectivity measures independent of those comorbid conditions. For the two rsfMRI measures with replicable associations with anhedonia, inclusion of the comorbid psychiatric conditions in the multiple linear regressions resulted in only a slight decrease in effect size and estimate of the partial regression coefficient for anhedonia. Together with the finding that none the partial regression coefficients for the comorbid psychiatric conditions were significantly associated with these two rsfMRI measures, the results suggest that the associations are specific to anhedonia.

The presence of anhedonia was associated with a decrease in the *Auditory vs. Right-Putamen* connectivity measure. Previous functional neuroimaging studies have found that functional activation between regions of the auditory cortex and putamen occurs during speech learning [40]. Specifically, higher coactivation was associated with incorrect categorization of auditory stimuli. The authors proposed that the putamen may act to "tune" activity in the auditory cortices to help facilitate learning how to correctly categorize tones that lead to positive

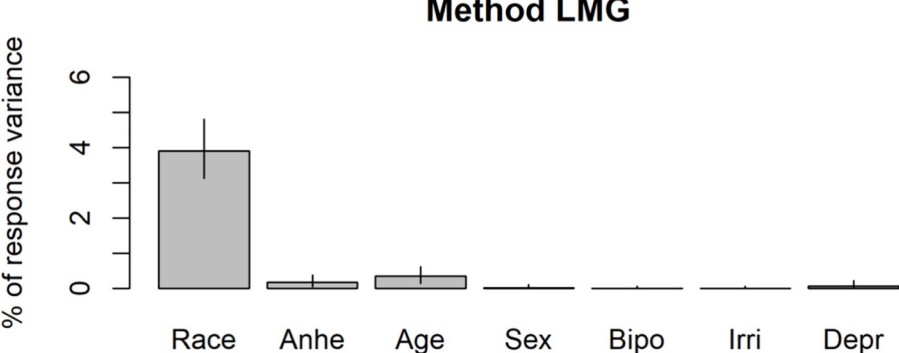

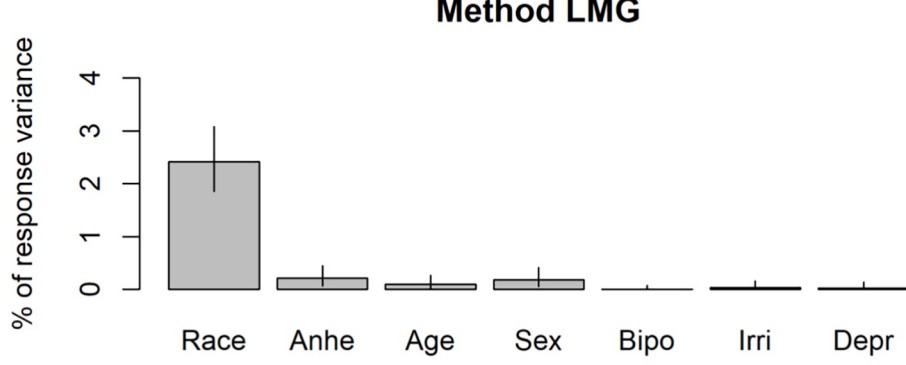

**Fig 2. Relative importance of independent variables in multiple linear regressions.** The proportion of the total variance in the Auditory vs. Right-Putamen (top) and Retrosplenial-Temporal vs. Right-Thalamus-Proper (bottom) rsfMRI measures accounted for by each of the independent variables in the regression models are shown. $R^2$ is the proportion of the total variance explained by each regression model. **Race** represents the race/ethnicity factor, **Anhe** represents anhedonia, **Age** represents age (in weeks), **Sex** represents participant sex, **Bipo** represents bipolar II disorder, **Irri** represents irritability, and **Depr** represents depressed mood.

feedback. In the context of anhedonia, reduced connectivity between auditory and the putamen may reflect a more general impairment in reward based learning [41]. For example, individuals with high levels of anhedonia have been shown to exhibit diminished ability to learn to modify their behavior during certain tasks in order to obtain larger rewards [42]. Decreased

connectivity between the striatal reward regions, such as the putamen, and cortical networks may disrupt the processes that drive behavioral and motivational adaptations that, in part, characterize anhedonia.

Anhedonia was also associated with decreases in the *Retrosplenial-Temporal vs. Right-Thalamus-Proper* connectivity measure. Few studies have reported on this connectivity measure in the context of anhedonia. The retrosplenial cortex has been associated with many cognitive processes, notably with impaired memory [43]. Interestingly, one study showed that individuals with social anhedonia exhibited increased functional connectivity between the retrosplenial cortex and several other brain regions, which were also correlated with lower anticipated pleasure in social situations that may point to the role of the retrosplenial cortex in future-oriented thinking [44]. As the thalamus has important functions in emotion and arousal [45], we speculate that a disrupted connectivity between retrosplenial networks and the thalamus may be associated with impaired future-oriented thinking about emotionally rewarding things or situations that may underlie the decreased motivational aspects of anhedonia. Interestingly, a recent study found significant anatomical connectivity between the restrosplenial cortex and fronto-striatal reward regions, suggesting a more direct involvement of the retrosplenial cortex in reward and decision making processes [46]. Altogether, our findings suggest that disrupted brain connectivity may underlie impairments in learning, emotional, and motivational processes associated with anhedonia. If brain dysfunction is a causal factor in the development of dysregulated processes associated with anhedonia, then they may become significant targets of preventative interventions and therapeutic treatments, including pharmacological or psychotherapies.

Our findings suggest that we have identified rsfMRI connectivity measures specifically associated with anhedonia that are independent from comorbid conditions, such as depressed mood. As such, the interpretation of anhedonia requires careful consideration and reflection. As mentioned previously, while anhedonia and depressed mood are core symptoms of major depressive disorders, they are considered distinct processes [17]. Alternatively, the hierarchical Taxonomy of Psychopathology (HiTOP) [47], a recently developed dimensional framework for psychopathology, has classified anhedonia as a symptom belonging to two high level *sepctra* of psychopathology; the *internalizing* and *detachment* spectra. Interestingly, low/depressed mood and irritability also fall under the *internalizing* spectrum whereas bipolar II falls under the *thought disorder* spectrum. Thus, when estimating the effects of anhedonia independent of other internalizing and thought disorder related symptoms, we could interpret the remaining effect to emphasize detachment processes. As detachment is a component of the psychosis super-spectrum [48] our findings may represent anhedonic neurocircuitry that may, in part, be related to schizophrenia, schizotypal personality, or other psychotic disorder processes. Further work is required to assess the extent to which differences in rsfMRI connectivity specific to anhedonia better associates with or even predicts dimensional measures of internalizing or detachment related psychopathology.

## Limitations

We acknowledge that while we referred to our analyses using the ABCD 4.0 (excluding 1.0) sample as analogous to replication in an independent sample, the participants do come from the same ABCD study and thus, our analyses do not conform to the strictest definition of a replication study. As such, shared aspects of data collection and processing, as well as other uncharacterized factors unique to the ABCD study may potentially impact our results. Future studies using a completely independent dataset will be required to assess the validity of our findings.

Another limitation of our study was the use of seed-based correlational methods to compute network connectivity measures in functionally-defined networks. As such, these network connectivity measures are averages over large and distributed networks where signals from sub-regions potentially highly associated with anhedonia may be drowned out by signals from sub-regions with low levels of association. Another concern is that the Gordon brain parcellations were produced using a boundary-mapping approach in adult brains [21] so whether they are generalizable to the brains of developing children is important to consider. For example, one study found that the functional topography of connectivity networks does change with age which was predictive of individual differences in executive function [49]. One alternative method is to use a decomposition-based method, such as independent components analysis (ICA), to define functional connectivity measures [50]. ICA is a data driven approach that extracts components that maximally explain the data and thus, may enhance predictive performance. One study took such an approach and found that using a decomposition-based, compared to a seed-based, extraction of functional networks during a social cognition task achieved significantly greater performance in predicting the degree of social anhedonia in around 70 adolescents/young adults with varying levels of schizotypy [51]. Since the raw neuroimaging data from the ABCD data are publicly available, this may be a feasible approach to implement in a future study.

In our study, we use a binary classifier for childhood anhedonia. However, several studies have reported greater predictive performance of neuroimaging measures on anhedonia symptom scores [51, 52] which suggests that functional neuroimaging measures may be more useful for predicting symptom severity rather than for disorder classification. Thus, it may be more reliable to investigate the associations between functional neuroimaging measures and clinical scales for assessing behavioral problems (such as the Child Behavior Checklist) or neurocognitive performance in individuals with anhedonia.

Finally, the DSM-V definition of anhedonia conflates two distinct reward processes: motivational (interest/wanting) and consummatory (pleasurable/liking) behaviors. These behaviors have been shown to have distinct neurobiological and behavioral components [10, 53]. We are limited in our study because we do not distinguish between these processes. However, the ABCD study data does include task-based functional neuroimaging of participants completing the monetary incentive delay task, which is able to assess the anticipatory, consummatory, and learning aspects of reward [54]. These processes were studied previously by Pornpattananangkul et al., but exceeded the scope of this study. Nevertheless, exploration of brain connectivity specific to each of these reward-based components in subjects with anhedonia may help elucidate the underlying circuitry underlying this complex psychiatric condition.

## Supporting information

**S1 Fig. Correlations between statistics from the current analyses and those reported by the previous authors.** Pearson correlations for student's t-statistics (left) and lnBF statistics (right) derived from the previous author's analyses and those derived from our reproduction analyses for **A)** anhedonia, **B)** depressed mood, and **C)** anxiety. LnBF are natural logarithm of Bayes Factors.
(DOCX)

**S2 Fig. Visual inspection of normality for rsfMRI connectivity measures from the ABCD 1.0 sample.** Density (left) and quantile-quantile (right) plots are shown for 3 representative rsfMRI connectivity measures.
(DOCX)

**S3 Fig. Comparing Welch's t-tests to Student's t-tests.** To assess the potential bias introduced by unequal variances across anhedonia and control groups, we performed Welch's t-tests for those rsfMRI measures with significant F-tests in the **A)** ABCD 1.0, **B)** ABCD 4.0 (excluding 1.0), and **C)** full ABCD 4.0 samples and then correlated the Welch's t-statistics with Student's t-statistics (left) and the Welch's p-values with Student's p-values (right).
(DOCX)

**S4 Fig. Visual inspection of normality for rsfMRI connectivity measures from the ABCD 4.0 (excluding ABCD 1.0) sample.** Density (left) and quantile-quantile (right) plots are shown for 3 representative rsfMRI connectivity measures.
(DOCX)

**S5 Fig. Visual inspection of normality for rsfMRI connectivity measures from the full ABCD 4.0 sample.** Density (left) and quantile-quantile (right) plots are shown for 3 representative rsfMRI connectivity measures.
(DOCX)

**S6 Fig. Visual inspection of residuals from the regression (controlling for sociodemographic covariates) for the *FrontoParietalRightVentraldc* rsfMRI connectivity measure using the ABCD 1.0 and correlations between OLS and WLS t-statistics.** Here, we visualize the residuals plotted against marginal fitted values of the linear regression model for the *FrontoParietalRightVentraldc* rsfMRI measure using the ABCD 1.0 sample (left), which exhibited a significant BP test for heteroskedasticity. We also performed weighted-least-squares regression (WLS) and correlated the t-statistics for each model predictor with those from the original ordinary-least-squares (OLS) regression (right).
(DOCX)

**S7 Fig. Visual inspection of residuals from the regression for *FrontoParietalRightVentraldc* rsfMRI measure (controlling for sociodemographic and psychiatric comorbidities) using the ABCD 1.0 sample and correlations between OLS and WLS t-statistics.** Here, we visualize the residuals plotted against marginal fitted values of the linear regression model for the *FrontoParietalRightVentraldc* rsfMRI measure using the ABCD 1.0 sample (left), which exhibited a significant BP test for heteroskedasticity. We also performed weighted-least-squares regression (WLS) and correlated the t-statistics for each of the model predictors with those from the original ordinary-least-squares (OLS) regressions (right).
(DOCX)

**S8 Fig. Visual inspection of residual variance from regressions (controlling for sociodemographic covariates only) using the ABCD 4.0 (excluding 1.0) sample for rsfMRI measures with significant Breusch-Pagan tests and correlations between OLS and WLS t-statistics.** Here, we visualize the residuals plotted against marginal fitted values for 4 rsfMRI connectivity measures exhibiting significant BP tests for heteroskedasticity. We also performed weighted-least-squares regression (WLS) for each rsfMRI measure and correlated the t-statistics for each of the model predictors with those from the original ordinary-least-squares (OLS) regressions. Note that some plots present a more discrete distribution of residuals along the fitted-values axis. These patterns occur when categorical predictors (such as race/ethnicity) exhibit greater effects in the model, compared to more continuous predictors (such as age), and thus are weighted more when producing predicted values.
(DOCX)

**S9 Fig. Visual inspection of residual variance from regressions (controlling for sociodemographic and psychiatric comorbidities) using the ABCD 4.0 (excluding 1.0) sample for**

**rsfMRI measures with significant Breusch-Pagan tests and correlations between OLS and WLS t-statistics.** Here, we visualize the residuals plotted against marginal fitted values for 4 rsfMRI connectivity measures exhibiting significant BP tests for heteroskedasticity. We also performed weighted-least-squares regression (WLS) for each rsfMRI measure and correlated the t-statistics for the model predictors with those from the original ordinary-least-squares (OLS) regressions.
(DOCX)

**S10 Fig. Visual inspection of residual variance from regressions (controlling for sociodemographic covariates only) using the full ABCD 4.0 sample for rsfMRI measures with significant Breusch-Pagan tests and correlations between OLS and WLS t-statistics.** Here, we visualize the residuals plotted against marginal fitted values for 4 rsfMRI connectivity measures exhibiting significant BP tests for heteroskedasticity. We also performed weighted-least-squares regression (WLS) for each rsfMRI measure and correlated the t-statistics for the model predictors with those from the original ordinary-least-squares (OLS) regressions.
(DOCX)

**S11 Fig. Visual inspection of residual variance from regressions (controlling for sociodemographic covariates and psychiatric comorbidities) using the full ABCD 4.0 sample for rsfMRI measures with significant Breusch-Pagan tests and correlations between OLS and WLS t-statistics.** Here, we visualize the residuals plotted against marginal fitted values for 4 rsfMRI connectivity measures exhibiting significant BP tests for heteroskedasticity. We also performed weighted-least-squares regression (WLS) for each rsfMRI measure and correlated the t-statistics for the model predictors with those from the original ordinary-least-squares (OLS) regressions.
(DOCX)

**S12 Fig. Visual inspection of normality for rsfMRI connectivity measure residuals from multiple linear regression models controlling for sociodemographic covariates using the ABCD 1.0 sample.** Density (left) and quantile-quantile (right) plots are shown for 3 representative rsfMRI connectivity measures with significant associations with anhedonia.
(DOCX)

**S13 Fig. Visual inspection of normality for rsfMRI connectivity measure residuals from multiple linear regression models with sociodemographic and psychiatric comorbidities using the ABCD 1.0 sample.** Density (left) and quantile-quantile (right) plots are shown for 3 representative rsfMRI connectivity measures with significant associations with anhedonia.
(DOCX)

**S14 Fig. Visual inspection of normality for rsfMRI connectivity measure residuals from multiple linear regression models with sociodemographic covariates using the ABCD 4.0 (excluding 1.0) sample.** Density (left) and quantile-quantile (right) plots are shown for 3 representative rsfMRI connectivity measures with significant associations with anhedonia.
(DOCX)

**S15 Fig. Visual inspection of normality for rsfMRI connectivity measure residuals from multiple linear regression models with sociodemographic covariates and psychiatric comorbidities using the ABCD 4.0 (excluding 1.0) sample.** Density (left) and quantile-quantile (right) plots are shown for 3 representative rsfMRI connectivity measures with significant associations with anhedonia.
(DOCX)

**S16 Fig. Visual inspection of normality for rsfMRI connectivity measure residuals from multiple linear regression models with sociodemographic covariates using the full ABCD 4.0 sample.** Density (left) and quantile-quantile (right) plots are shown for 3 representative rsfMRI connectivity measures with significant associations with anhedonia.
(DOCX)

**S17 Fig. Visual inspection of normality for rsfMRI connectivity measure residuals from multiple linear regression models with sociodemographic covariates and psychiatric comorbidities using the full ABCD 4.0 sample.** Density (left) and quantile-quantile (right) plots are shown for 3 representative rsfMRI connectivity measures with significant associations with anhedonia.
(DOCX)

**S1 Table. T-test results comparing those with anhedonia from controls using the ABCD 1.0 sample.** Effect sizes (Cohen's d) with 95% Cis, t-statistics, p-values, ln(Bayes Factor) (lnBF), group means with standard deviations (SD), Shapiro-Wilk statistics (W-statistic), and F-statistics are reported.
(XLSX)

**S2 Table. T-test results comparing those with anhedonia from controls using the ABCD 4.0 (excluding 1.0) sample.** Effect sizes (Cohen's d) with 95% Cis, t-statistics, p-values, ln (Bayes Factor) (lnBF), group means with standard deviations (SD), Shapiro-Wilk statistics (W-statistic), and F-statistics are reported.
(XLSX)

**S3 Table. T-test results comparing those with anhedonia from controls using the full ABCD 4.0 sample.** Effect sizes (Cohen's d) with 95% Cis, t-statistics, p-values, ln(Bayes Factor) (lnBF), group means with standard deviations (SD), Shapiro-Wilk statistics (W-statistic), and F-statistics are reported.
(XLSX)

**S4 Table. Multiple linear regression results in the ABCD 1.0 sample, controlling for socio-demographic covariates.** The partial regression coefficients (Estimate), standard errors (Std. Err), t-values, p-values, significance, model R2 (R^2), Bonferroni-Hochberg Adjusted p-values (BH adjustment), Bonferroni adjustd p-values, Durbin-Watson statistic (DW_statistic), Breusch-Pagan Chi^2 (BP Chi^2), and Breusch-Pagan (BP) p-values are presented. The ICC is the proportion of variance in rsfMRI connectivity explained by the family structure random effect.
(XLSX)

**S5 Table. Multiple linear regression results in ABCD 1.0 sample, controlling for sociode-mographic covariates and psychiatric comorbidities.** The partial regression coefficients (Estimate), standard errors (Std.Err), t-values, p-values, significance, model R2 (R^2), Bonfer-roni-Hochberg Adjusted p-values (BH adjustment), Bonferroni adjustd p-values, Durbin-Watson statistic (DW_statistic), Breusch-Pagan Chi^2 (BP Chi^2), and Breusch-Pagan (BP) p-values are presented. The ICC is the proportion of variance in rsfMRI connectivity explained by the family structure random effect.
(XLSX)

**S6 Table. Multiple linear regression results in ABCD 4.0 (excluding 1.0) sample, control-ling for sociodemographic covariates.** The partial regression coefficients (Estimate), standard errors (Std.Err), t-values, p-values, significance, model R2 (R^2), Bonferroni-Hochberg

Adjusted p-values (BH adjustment), Bonferroni adjustd p-values, Durbin-Watson statistic (DW_statistic), Breusch-Pagan Chi^2 (BP Chi^2), and Breusch-Pagan (BP) p-values are presented. The ICC is the proportion of variance in rsfMRI connectivity explained by the family structure random effect.
(XLSX)

**S7 Table. Multiple linear regression results in ABCD 4.0 (excluding 1.0) sample, controlling for sociodemographic covariates and psychiatric comorbidities.** The partial regression coefficients (Estimate), standard errors (Std.Err), t-values, p-values, significance, model R2 (R^2), Bonferroni-Hochberg Adjusted p-values (BH adjustment), Bonferroni adjustd p-values, Durbin-Watson statistic (DW_statistic), Breusch-Pagan Chi^2 (BP Chi^2), and Breusch-Pagan (BP) p-values are presented. The ICC is the proportion of variance in rsfMRI connectivity explained by the family structure random effect.
(XLSX)

**S8 Table. Multiple linear regression results in the full ABCD 4.0 sample, controlling for sociodemographic covariates.** The partial regression coefficients (Estimate), standard errors (Std.Err), t-values, p-values, significance, model R2 (R^2), Bonferroni-Hochberg Adjusted p-values (BH adjustment), Bonferroni adjustd p-values, Durbin-Watson statistic (DW_statistic), Breusch-Pagan Chi^2 (BP Chi^2), and Breusch-Pagan (BP) p-values are presented. The ICC is the proportion of variance in rsfMRI connectivity explained by the family structure random effect.
(XLSX)

**S9 Table. Multiple linear regression results in the full ABCD 4.0 sample, controlling for sociodemographic covariates and psychiatric comorbidities.** The partial regression coefficients (Estimate), standard errors (Std.Err), t-values, p-values, significance, model R2 (R^2), Bonferroni-Hochberg Adjusted p-values (BH adjustment), Bonferroni adjustd p-values, Durbin-Watson statistic (DW_statistic), Breusch-Pagan Chi^2 (BP Chi^2), and Breusch-Pagan (BP) p-values are presented. The ICC is the proportion of variance in rsfMRI connectivity explained by the family structure random effect.
(XLSX)

**S10 Table. Comparison of participants removed due to missing data with participants who were retained for the t-test analyses.** The proportion (%) of participants in each group for each measure are shown. Student's t-test was done to compare age (in months) between the "removed" and "retained" groups. Chi-square tests of independence were done for all other measures. P-values < 0.05 are bolded.
(XLSX)

**S11 Table. Tetrachoric correlations between youth reported anhedonia and other psychiatric conditions reproted by parents or youths.** Items ending in "_y" are youth reported measures, and items ending in "_p" are parent reported measures.
(XLSX)

**S12 Table. Prevalence of significantly comorbid psychiatric symptoms and diagnoses in those reporting anhedonia at baseline.** The frequencies of three psychiatric symptoms and diagnoses, in individuals reporting anhedonia, significantly correlated (tetrachoric correlation r > = 0.5) with youth reported anhedonia are shown. Note that before filtering out participants with low-quality functional MRI scans, 1104 participants at baseline reported past and/or present anhedonia. The suffix "_y" indicate measures based on youths' self-reports.
(XLSX)

**S13 Table. T-tests results for comparisons between those with depressed mood, irritability, and bipolar II disorder and controls using the full ABCD 4.0 sample.** Effect sizes (Cohen's d) with 95% Cis, t-statistics, p-values, and ln(Bayes Factor) (lnBF) statistics are reported. Bolded values represent statistically significant results, as indicated by lnBF values > 1.1. (XLSX)

**S14 Table. Average Generalized Variance Inflation Factor (GVIF) values for predictors from multiple linear regressions controlling for sociodemographic covariates across the ABCD samples.**
(XLSX)

**S15 Table. Average Generalized Variance Inflation Factor (GVIF) values for predictors from multiple linear regressions controlling for sociodemographic covariates and psychiatric comorbidities across the ABCD samples.**
(XLSX)

**S16 Table. Comparison of participants removed due to missing data with participants who were retained for the multiple linear regression analyses.** The proportion (%) of participants in each group for each measure are shown. Student's t-test was done to compare age (in months) between the "removed" and "retained" groups. Chi-square tests of independence were done for all other measures. p-values < 0.05 are bolded. (XLSX)

**S17 Table. Effects of bipolar II disorder, irritability, and depressed mood from multiple linear regression analyses across the ABCD study samples for 27 rsfMRI measures associated with anhedonia.** Effect size (effect_CIs) with 95% confidence intervals represents the percentage of the total variance (proportion of variance * 100%) in a rsfMRI connectivity measure accounted for by each predictor (psychiatric symptom or diagnosis). Total model R^2 (total_R2), partial regression coefficients (Estimate), standard errors (Std.Err), t-values, p-values, and Benjamini-Hochberg corrected p-values (BH_adjustment) values are reported. (XLSX)

## Acknowledgments

Data used in the preparation of this article were obtained from the Adolescent Brain Cognitive Development[SM] (ABCD) Study (https://abcdstudy.org), held in the NIMH Data Archive (NDA). This is a multisite, longitudinal study designed to recruit more than 10,000 children age 9–10 and follow them over 10 years into early adulthood. The ABCD Study® is supported by the National Institutes of Health and additional federal partners under award numbers U01DA041048, U01DA050989, U01DA051016, U01DA041022, U01DA051018, U01DA051037, U01DA050987, U01DA041174, U01DA041106, U01DA041117, U01DA041028, U01DA041134, U01DA050988, U01DA051039, U01DA041156, U01DA041025, U01DA041120, U01DA051038, U01DA041148, U01DA041093, U01DA041089, U24DA041123, U24DA041147. A full list of supporters is available at https://abcdstudy.org/federal-partners.html. A listing of participating sites and a complete listing of the study investigators can be found at https://abcdstudy.org/consortium_members/. ABCD consortium investigators designed and implemented the study and/or provided data but did not necessarily participate in the analysis or writing of this report. This manuscript reflects the views of the authors and may not reflect the opinions or views of the NIH or ABCD consortium investigators.

The ABCD data repository grows and changes over time. The ABCD data used in this report came from http://dx.doi.org/10.15154/1523041.

This manuscript reflects the views of the authors and may not reflect the opinions or views of the NIH or ABCD consortium investigators.

This study received no external funding. Yi Zhou was supported by the department of Psychiatry at Virginia Commonwealth University.

## Author Contributions

**Conceptualization:** Yi Zhou.

**Formal analysis:** Yi Zhou.

**Investigation:** Yi Zhou.

**Methodology:** Yi Zhou.

**Resources:** Narun Pat.

**Supervision:** Michael C. Neale.

**Writing – original draft:** Yi Zhou.

**Writing – review & editing:** Yi Zhou, Narun Pat, Michael C. Neale.

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
