## [Decision Letter · Decision Letter 0]

19 Dec 2022

PONE-D-22-28899Associations Between Resting State Functional Brain Connectivity and Childhood Anhedonia: A Reproduction and Replication StudyPLOS ONE

Dear Dr. Zhou,

Thank you for submitting your manuscript to PLOS ONE. After careful consideration, we feel that it has merit but does not fully meet PLOS ONE’s publication criteria as it currently stands. Therefore, we invite you to submit a revised version of the manuscript that addresses the points raised during the review process.

We look forward to receiving your revised manuscript.

Kind regards,

Melissa A Brotman, PhD

Academic Editor

PLOS ONE

Journal Requirements:

2. Our internal editors have looked over your manuscript and determined that it is within the scope of our Reproducibility and Replicability in Neuroscience and Mental Health Research Call for Papers. The Collection will encompass a diverse and interdisciplinary set of protocols and research articles adhering to transparent and reproducible reporting practices in the areas of clinical psychology, psychiatry, mental health, and neuroscience. Additional information can be found on our announcement page: https://collections.plos.org/call-for-papers/reproducibility-and-replicability-in-neuroscience-and-mental-health-research/. If you would like your manuscript to be considered for this collection, please let us know in your cover letter and we will ensure that your paper is treated as if you were responding to this call. If you would prefer to remove your manuscript from collection consideration, please specify this in the cover letter.

Reviewers' comments:

Reviewer's Responses to Questions

**Comments to the Author**

1. Is the manuscript technically sound, and do the data support the conclusions?

Reviewer #1: Yes

Reviewer #2: Yes

2. Has the statistical analysis been performed appropriately and rigorously? 

Reviewer #1: Yes

Reviewer #2: Yes

3. Have the authors made all data underlying the findings in their manuscript fully available?

Reviewer #1: Yes

Reviewer #2: Yes

4. Is the manuscript presented in an intelligible fashion and written in standard English?

Reviewer #1: Yes

Reviewer #2: Yes

5. Review Comments to the Author

Reviewer #1: The present study sought to test reproducibility and replicability of findings regarding relations between anhedonia and resting state functional connectivity in 9 and 10 year olds. Leveraging data from two releases of the ABCD study, the authors replicate a limited subset of findings from previously published work. Reproducibility and replicability studies are necessary in the fields of psychology and neuroscience, yet they often go unpublished highlighting the merit of dissemination of these findings. The manuscript would be strengthened by further elaboration on the potential implications for assessment and treatment of anhedonia and depression which is missing from the discussion. Additional suggestions and questions are detailed:

Introduction

1. In the abstract, the phrase 'children reporting anhedonia' is used to describe the sample. Further reading of the manuscript clarifies that a binary variable was used to classify children as endorsing or not endorsing anhedonia. Additionally, the methods seem to imply that both parent and child report was used to define whether or not youth had anhedonia. This should be stated more clearly in the abstract, perhaps as 'measured associated with presence of anhedonia based on parent and child reports.'

2. In the introduction, can the authors please clarify whether anhedonia in children and adolescents is a significant predictor of greater depression severity, treatment resistant depression, and suicidal behaviors cross-sectionally, longitudinally, or both?

3. Functional connectivity may also reflect co-deactivation or deactivation/activation that occurs simultaneously in brain regions. Therefore, the section 'the coactivation of different brain regions, which measures' should be cut from lines 72-73 of the introduction.

4. The authors remark the importance of functional connectivity being able to be measured at rest in the introduction. Why is this important and in the context of the present study, what would be the benefit of looking at resting state functional connectivity as opposed to task based?

5. Per the definition of replicability in lines 92-94, I don't think this study would constitute as full/true reproducibility given that this is not really a different dataset--just different participants in the same dataset. This should be noted in the limitations.

6. Study aims are lacking the why--why do we need to understand brain dysfunction underlying anhedonia?

Methods

7. The authors used a recent version of R--is this the same version that was used in the original manuscript they are reproducing/replicating? Different versions of statistical softwares and even computer softwares have been shown to yield different results. The authors should note whether there were software differences, and perhaps consider re-running the analyses on the same version(s) to compare.

8. Were the individuals scanned by the Philips Medical Systems MRI machines in ABCD 1.0 release included in the original manuscript that is being replicated/reproduced? Please clarify, and please clarify if this resulted in a difference in sample sizes between the two papers.

9. How were the parent and child reports combined for use in analyses?

10. How were variables quantified? Were the items for depressed mood, irritability, and anhedonia yes/no response items, or was a threshold set to define presence of the symptom based on a continuous response?

11. How many participants with missing data were removed? What was the pattern of missing data? Were there any significant differences in study variables or demographic individuals in those with/without missing data?

12. What was the amount of outliers corrected?

13. How was normality examined and, if necessary, were any data transformations performed?

14. Please report on model assumptions for multiple linear regression and how any violations were handled.

15. How many families were in the study? ICC values for the nesting factor?

Results

16. Line 237--what findings?

17. Lines 247-249: only 4 connectivity measures survived correction for multiple comparisons. How does this map onto the findings of the original study after correcting for multiple comparisons?

18. Tables 2-4 are cut off. Do the tables include effect sizes and confidence intervals? If not, please include.

19. Given the number of comparisons made and the sample size, the risk of false positives is high, and only findings that survive multiple comparisons should be interpreted.

20. The authors make a number of comments about statistical power. Given the sample size, there is sufficient power to detect significant effects if they exist, and in fact the risk of false positives is increased in this larger sample. Rather, the more informative measure in such a large sample would be the magnitude of that effect. Effect sizes should be reported and interpreted for all findings, as well as confidence intervals.

21. In table 4, is 'Effect of Anhedonia' supposed to be the B value?

22. R2 values are quite small--effects are extremely small. Authors should note this as a limitation--effect does not seem to be meaningful.

23. What was the n for the multiple regression analyses? I don't think a sample of thousands can be considered 'relatively small' as stated in lines 378 and 447.

Discussion

24. The meaning of the networks identified are not really elaborated on in the discussion, nor are clinical implications at least speculated. I might recommend the authors putting the knowledge expanded on in this study into clinical context.

Minor:

- One citation in the introduction (line 68) is in a different format compared to all other citations

- 'prognostic predictor' in line 69 of the introduction is redundant

- In line 128, ROIs should be plural, not possessive (ROI's)

- Line 225, the two groups don't' appear to exhibit', they DO statistically exhibit differences.

Reviewer #2: The authors report on reproducibility and replicability of the Pornpattananangkul 2019 JAMA results on associations between alterations in rsfMRI connectivity measures and anhedonia in youth.

This is important work that I would like to see published, provided some significant revisions to the manuscript.

My main concern is that I didn’t feel like I got a good understanding what was and what was not reproduced/replicated. Part of the issue is that the results were worded much differently in the original paper, with a focus on directionality of differences. I would very much like to see a summary table that provides a summary of sets of findings directly compared. It’s an arduous process to go through and double check that indeed all differences are reproduced/replicated. I feel particularly strong about this point in that statistical thresholding is always to some extent arbitrary – seeing results with p values displayed in parallel may help illustrate what does and what does not replicate.

It was not clear to me that all analyses from the original rsfMRI paper were reproduced. What about the analyses showing specificity?

I find it curious that the paper seems to suggest that the analyses were run by a team of independent researchers when the first author of the initial paper is the second author of the current paper if I'm not mistaken. It’s not an issue per se, but I want to make sure that all coding for the reproducibility analyses was done from scratch, otherwise it may just carry forward errors.

May be worth presenting the correlations between covariates and anhedonia in the supplements. How were covariates decided on? Why was trauma or anxiety not included for example?

Partial regression coefficients of all significant predictors should be displayed, not just the rsFMRI measures.

The results from the new specificity analyses were not nearly discussed enough in the discussion section nor was the effect of controlling for additional covariates on the 1.0 sample.

The result section in the abstract should explicitly state that the larger full sample that replicated 6/11 associations included the data from the initial study and is therefore not an independent replication. In the same vein, the wording “replication of previous findings was limited” maybe overly optimistic given that in the independent sample only 1/11 was replicated, and, once controlled for covariates, a lot of the associations in the initial sample were no longer significant, too.

Generally, the writing was clear, and I appreciate the effort that goes into this kind of work – it really serves the field.

6. PLOS authors have the option to publish the peer review history of their article (what does this mean?). If published, this will include your full peer review and any attached files.

Reviewer #1: No

Reviewer #2: No

---

## [Author Response · Author response to Decision Letter 0]

7 Feb 2023

Reviewer #1

Introduction

1. In the abstract, the phrase 'children reporting anhedonia' is used to describe the sample. Further reading of the manuscript clarifies that a binary variable was used to classify children as endorsing or not endorsing anhedonia. Additionally, the methods seem to imply that both parent and child report was used to define whether or not youth had anhedonia. This should be stated more clearly in the abstract, perhaps as 'measured associated with presence of anhedonia based on parent and child reports.'

[Response] Participants with anhedonia were indeed classified based on the youth reports only, which is the same as the approach the previous authors took. Parent-reported measures were shown to cluster very distinctly from the youth-reported measures in our correlation matrix of youth and parent measures (Figure. 1), indicating a high level of discordance between youth and parent reported measures. We clarified in the methods how we used the youth and parent reported measures (see response to reviewer 1, comment 9). 

2. In the introduction, can the authors please clarify whether anhedonia in children and adolescents is a significant predictor of greater depression severity, treatment resistant depression, and suicidal behaviors cross-sectionally, longitudinally, or both?

[Response] We have revised our introduction which now reads:

“In cross-sectional studies, anhedonia in children and adolescents has been shown to be associated with greater depression severity and suicidality (7,8). Furthermore, in a randomized clinical trial, anhedonia was found to be a significant predictor of a longer time to remission in adolescents with treatment resistant depression (9).” (lines 70-37)

3. Functional connectivity may also reflect co-deactivation or deactivation/activation that occurs simultaneously in brain regions. Therefore, the section 'the coactivation of different brain regions, which measures' should be cut from lines 72-73 of the introduction.

[Response] We have removed the specified lines from the introduction which now reads:

“Functional brain connectivity is a measure of the degree of synchrony between the blood oxygen level dependent (BOLD) signals across time between regions in the brain (Lv et al., 2018).” (lines 76-78)

4. The authors remark the importance of functional connectivity being able to be measured at rest in the introduction. Why is this important and in the context of the present study, what would be the benefit of looking at resting state functional connectivity as opposed to task based?

[Response] We have revised the introduction which now reads:

“Importantly, functional connectivity can be measured at rest which facilitates the ease by which data can be collected, as opposed to task-fMRI which typically involves the presence of a stimulus or task (11). While task-fMRI focuses on patterns of brain activation associated with a specific task, resting-state functional MRI (rsfMRI) connectivity focuses on the organization of brain networks that may specialize in specific functions. As such, both task-fMRI and rsfMRI represent important brain processes that may contribute to the development of mental disorders” (lines 80-86)

5. Per the definition of replicability in lines 92-94, I don't think this study would constitute as full/true reproducibility given that this is not really a different dataset--just different participants in the same dataset. This should be noted in the limitations.

[Response] We acknowledge this limitation and have included the following lines in the limitations section of the discussion: 

“We acknowledge that while we referred to our analyses using the ABCD 4.0 (excluding 1.0) sample as analogous to replication in an independent sample, the participants do come from the same ABCD study and thus, our analyses do not conform to the strictest definition of a replication study. As such, shared aspects of data collection and processing, as well as other uncharacterized factors unique to the ABCD study may potentially impact our results. Future studies using a completely independent dataset will be required to assess the validity of our findings.” (lines 825-831)

6. Study aims are lacking the why--why do we need to understand brain dysfunction underlying anhedonia?

[Response] We have revised the introduction which now reads:

“Anhedonia is a transdiagnostic symptom impacting some of the most prevalent and debilitating mental disorders in both children and adults. By Identifying brain dysfunction underlying anhedonia, we may be able to more accurately characterize patient symptomatology, refine diagnostic categories, and potentially identify specific brain-based therapeutic targets. Furthermore, by characterizing brain dysfunction in children and adolescents, we may help elucidate important dysregulated developmental processes that may be targeted by preventative interventions.” (lines 116-122)

Methods

7. The authors used a recent version of R--is this the same version that was used in the original manuscript they are reproducing/replicating? Different versions of statistical softwares and even computer softwares have been shown to yield different results. The authors should note whether there were software differences, and perhaps consider re-running the analyses on the same version(s) to compare.

[Response] For our manuscript, we used R version 4.0.3 (2020-10-10) while the previous authors used R version 3.4.3 (November 2017). However, there were no differences between our results and the previous authors’. We previously demonstrated this by correlating the t-statistics (as well as lnBF statistics) from our reproduced associations between rsfMRI connectivity and anhedonia and those from the previous authors’ analyses and show a correlation r = 1 (previously Figure 1A and 1B), indicating the results were exactly the same. Of note, the previous authors’ also analyzed the association between rsfMRI connectivity and depressed mood, as well as with anxiety, to assess the specificity of their anhedonia results. Thus, we have now also reproduced those same associations and include the correlations between the t-statistics (and lnBF statistics) from our results and the previous authors’ to further demonstrate the equivalence of our results. The correlations were all r = 1, indicating the results were exactly the same. Plots of those correlations can now be found in Supplementary Figure 1. Furthermore, we have revised our manuscript to include the following lines:

“We note that although we used a more recent version of the R statistical software than the previous authors, the correlations between the statistics from our analyses and those from the previous authors’ were all equal to 1, indicating no differences in the results.” (lines 341-344)

8. Were the individuals scanned by the Philips Medical Systems MRI machines in ABCD 1.0 release included in the original manuscript that is being replicated/reproduced? Please clarify, and please clarify if this resulted in a difference in sample sizes between the two papers.

[Response] Individuals scanned by Philips Medical Systems MRI machines were also excluded from the previous authors’ manuscript. Thus, there were no differences in the sample sizes used in our analyses to reproduce the previous authors findings using data from the ABCD 1.0 data release. We have edited our manuscript which now reads:

“For analyses using only the ABCD 1.0 release sample, like the previous authors, we also removed individuals who were scanned by “Philips Medical Systems” MRI machines because of a post-processing issue in the ABCD 1.0 release, which was resolved in later releases.” (lines 160-163) 

“Like the previous authors, we identified 215 individuals who endorsed past and/or present anhedonia and 2,222 controls who reported neither past nor present anhedonia at the baseline timepoint, indicating the samples were exactly the same.” (lines 306-308)

However, we also used ABCD 1.0 data (excluding Philips Medical Systems participants) in our first multiple linear regression analysis that controlled for demographic covariates. Due to listwise deletion of missing data that included these new covariates, 13 participants were removed from the control group due to missingness in these new covariates. As such, the sample size differed from our reproduction t-test analysis. These differences were originally noted in the table legends for tables comparing demographic variables. However, we have edited our manuscript that now reads: 

“We note that participants with missing covariate data were excluded from these analyses. Thus, the sample sizes will differ slightly from those used in the previous t-tests. In the ABCD 1.0 release sample, there were 215 individuals who endorsed anhedonia and 2209 controls who did not.” (lines 535-539) 

9. How were the parent and child reports combined for use in analyses?

[Response] Parent and child reported questionnaire items were combined separately. We have clarified this in the manuscript which now reads:

“Separately for youth and parent KSADS questionnaire items, we combined past and present items for the same symptom or diagnosis and consolidated some items into a single variable. For example, we consolidated 18 youth-reported past and present suicide related diagnosis items into one single youth-reported suicide thoughts and behavior variable, which was similarly done in another study (21). A separate parent-reported suicide thoughts and behaviors variable was constructed using parent-reported KDADS questionnaire items.” (lines 186-192)

10. How were variables quantified? Were the items for depressed mood, irritability, and anhedonia yes/no response items, or was a threshold set to define presence of the symptom based on a continuous response?

[Response] All the psychiatric variables from the KSADS youth and parent questionnaires were binary yes/no variables. We have clarified this in the manuscript which now reads:

“Psychiatric data were obtained from the youth (data structure: abcd_ksad501) and parent (data structure: abcd_ksad01) Kiddie Schedule for Affective Disorders and Schizophrenia (KSADS) data structures from the ABCD study which are composed of binary (yes/no) questionnaire items for various psychiatric diagnoses or symptoms.” (lines 182-186)

11. How many participants with missing data were removed? What was the pattern of missing data? Were there any significant differences in study variables or demographic individuals in those with/without missing data?

[Response] For t-test analyses, consistent with the previous authors, study participants with missing rsfMRI were removed prior to adjustments for site using the ComBat tool. Additionally, participants with missing data for anhedonia were also removed. For the t-test analyses in the ABCD 1.0 sample, 288 participants were removed and 2437 were retained. The only statistically significant difference was that the participants who were removed were on average 1 week younger than those who were retained (Supplementary Table. 10). Such a difference was deemed to be negligible. 

 For the t-test analyses in the ABCD 4.0 (excluding 1.0) sample, 126 participants were removed and 6454 participants were retained. We believe less participants were removed than in the ABCD 1.0 sample because the ABCD 4.0 sample was a later release with more complete rsfMRI data. However, the participants who were removed had a significantly higher proportion of African American individuals (27% vs 16%) and a higher proportion of individuals with depressed mood (13% vs 9%) (Supplementary Table. 10). While significant patterns of missingness suggest Missingness At Random (MAR), the proportion of missing participants was ~2% and would not benefit from multiple imputation (missingness >5% would warrant multiple imputation) and are unlikely to lead to significant bias in results (1). Similarly, for the t-test analyses in the full ABCD 4.0 sample, 139 participants were removed and 8866 were retained (Supplementary Table. 10). Again, there were higher proportions of African American individuals and those with depressed mood in the participants who were removed. However, the proportion of participants removed was ~ 1.5% of the total sample and thus unlikely to contribute to significant bias to the analyses or receive benefit from multiple imputation. 

 For the multiple linear regression analyses, participants with missing demographic covariate data were also removed. However, this only led to a few more participants being excluded for each analysis (Supplementary Table. 16). Patterns of missingness were also compared and closely resembled those identified above. 

To reflect these important considerations, we have added the following additional section to the methods:

“Assessing Patterns of Missingness

For t-test analyses, consistent with the previous authors, study participants with missing rsfMRI measures were removed prior to adjustments for site using the ComBat tool. Additionally, participants with missing data for youth-reported anhedonia were also excluded. To assess if there were any patterns of missingness, we compared participants who were removed with participants who were retained. We assessed whether there were differences in proportions for sex (assigned at birth), race/ethnicity, youth-reported anhedonia, depressed mood, irritability, and bipolar II disorder symptoms between the two groups using Chi-Square tests of independence. We compared differences in age (in weeks) using t-tests. 

 For the multiple linear regression analyses, in addition to participants with missing rsfMRI and anhedonia responses, those with missing demographic covariate data were also excluded. Similar comparisons were made between participants who were removed and those who were retained as those reported above.” (lines 276-285)

Additionally we have added the following lines to the results section of the manuscript:

“For these t-tests using the ABCD 1.0 sample, 288 participants with either missing rsfMRI data or youth-reported anhedonia responses were removed and 2437 were retained. Participants who were removed were on average 1 week younger than those who were retained (Supplementary Table. 10). Such a difference was deemed to be negligible.” (lines 366-370)

“For these t-tests using the ABCD 4.0 (excluding ABCD 1.0) sample, 124 participants with either missing rsfMRI or youth-reported anhedonia responses were excluded and 6456 participants were retained. Notably, those that were removed exhibited significantly higher proportions of individuals identifying as African Americans (27.4% vs 15.7%) and individuals endorsing depressed mood (12.9% vs. 8.7%) (Supplementary Table. 10). While significant patterns of missingness suggest Missingness At Random (MAR), the proportion of missing participants was ~2% and thus would not benefit from multiple imputation (typically requiring missingness >5%) and are unlikely to lead to significant bias in the results (22).” (lines 400-408)

“For these t-tests using the full ABCD 4.0 sample, 139 participants with either missing rsfMRI or youth-reported anhedonia responses were removed and 8866 participants were retained. Similarly, there were higher proportions of individuals identifying as African American and those endorsing depressed mood in the participants who were removed. However, the proportion of participants removed was ~ 1.5% of the total sample and thus unlikely to contribute to significant bias to the analyses or receive benefit from multiple imputation.” (lines 438-444).

“For multiple linear regression analyses, patterns of missingness closely resembled those reported for the samples used in the t-tests. Notably, individuals removed due to missingness exhibited higher proportions of those identifying as African American and those endorsing depressed mood (Supplementary Table. 16). However, the proportion of participants removed was less than 5% of the total sample for each set of analyses and thus, were unlikely to bias the results.” (lines 643-648)

12. What was the amount of outliers corrected?

[Response] We have now included the number of outliers removed for each of the analyses in the corresponding supplementary tables (ex. Supplementary Table 1-9). On average, the proportion of the total sample removed as outliers was around 5% in the ABCD 1.0 sample, around 1.8% in the ABCD 4.0 (excluding 1.0) sample, and 1.6% in the full ABCD 4.0 sample. We have no included the following lines in the methods of our manuscript:

“The number and the percentage of the total sample removed as outliers for each t-test were reported in Supplementary Tables. 1-3)” (lines 208-209)

“Similar to the t-tests, prior to statistical analyses, participants with missing data for psychiatric symptoms/diagnoses, as well as demographic covariates, were removed. We then proceeded to identify and remove outliers for each rsfMRI connectivity measure as values 1.5 times greater than the interquartile range (IQR) of values. The number and the percentage of the total sample removed as outliers for each multiple linear regression were reported in Supplementary Tables. 4-9.” (lines 238-243)

13. How was normality examined and, if necessary, were any data transformations performed?

[Response] For large samples, such as in the ABCD study, statistical tests for normality (such as the Shapiro-Wilk test) are very sensitive to small deviations from normality which do not end up affecting the results of parametric tests. Furthermore, the central limit theorem states that for samples with n > 40, the means of random samples from any distribution tends to be normal, regardless of the distribution of the underlying data (2). As such, visual inspection of the distribution of variables is sufficient to detect any major deviations from normality. Nevertheless, we have now included normality tests for rsfMRI measures from the ABCD samples using the Shapiro-Wilk test and have included their results in corresponding supplementary tables for the t-test results (Supplementary Table 1-3). We have added the following lines to the manuscript:

“To check the assumptions for independent measures t-tests, we performed Shapiro-Wilk normality tests for rsfMRI measures from the ABCD 1.0 sample and found that all of the tests were significant, indicating non-normality (Supplementary Table. 1). However, for large samples, such as in the ABCD study, statistical tests for normality are very sensitive to small deviations from normality which do not end up affecting the results of parametric tests. Furthermore, the central limit theorem states that for samples with n > 40, the means of random samples from any distribution tends to be normal, regardless of the distribution of the underlying data (25). A visual inspection of the distributions, as well as the quantile-quantile (QQ) plots, for a representative sample of 3 of the rsfMRI measures significantly associated with anhedonia revealed no significant deviations from normality (Supplementary Figure 2).” (lines 345-355)

Shapiro-Wilk normality tests for rsfMRI measures from the ABCD 4.0 (excluding 1.0) sample and the full ABCD 4.0 sample could not be performed as the shapiro.test() function in R cannot be applied to samples > 5000, which are large enough such that the assumption of normality is not required due to the central limit theorem. Nevertheless, we visually inspected a representative sample of 3 rsfMRI measures significantly associated with anhedonia which revealed no significant deviations from normality (Supplementary Figures 4-5). We have included the following lines in the manuscript:

“To check the assumptions for independent samples t-tests performed using the ABCD 4.0 (excluding 1.0) and full ABCD 4.0 samples, we visually inspected the distributions of a representative sample of 3 rsfMRI measures significantly associated with anhedonia in each ABCD sample, which revealed no significant deviations from normality (Supplementary Figures 4 and 5).” (lines 445-449)

Another assumption of independent samples t-tests is that of homogeneity of variance. Thus, we have performed F-tests to compare variances across the anhedonia and control groups for our t-tests in order to assess the assumptions of homoscedasticity. We note that there were 11 rsfMRI measures that exhibited significant F-tests in the ABCD 1.0 sample, indicating we cannot assume equal variances across groups for those measures (Supplementary Table 1.). For these measures, non-parametric tests that do not assume homoscedasticity, such as Welch’s t-test, could be used to obtain more reliable results. Similarly, there were 17 rsfMRI measures in the ABCD 4.0 (excluding ABCD 1.0) sample, and 21 rsfMRI measures in the full ABCD 4.0 sample that also exhibited significant F-tests. To assess the potential bias introduced by unequal variances, we performed Welch’s t-tests for those rsfMRI measures with significant F-tests and then correlated the Welch’s t-statistics with Student’s t-statistics. We then correlated the p-values from the two tests as well. We found significantly high correlations (r >= 0.99) for all the correlations, indicating minimal differences between the two types of tests and thus minimal impact of unequal variances on the results (Supplementary Figure. 3). We have added the following lines to the manuscript:

“We also performed F-tests to compare variances across the anhedonia and control groups for our t-tests in order to assess the assumptions of homoscedasticity. We note that there were 11 rsfMRI measures that exhibited significant F-tests in the ABCD 1.0 sample (Supplementary Table. 1.). To assess the potential bias introduced by unequal variances, we performed Welch’s t-tests for those rsfMRI measures with significant F-tests and then correlated the Welch’s t-statistics with Student’s t-statistics. We then correlated the p-values from the two tests as well. We found significantly high correlations (r >= 0.99) for t-statistics and p-values, indicating minimal differences between the two types of tests and thus minimal impact of unequal variances on the results (Supplementary Figure. 3A).” (lines 356-365)

Finally, we have made the following revisions to the methods which now reads:

“Assumptions of normality were assessed with the Shapiro-Wilk test and with the visual inspection of the distributions of a representative sample of rsfMRI variables. Assumptions of equal variances across groups were assessed with the F-test. To assess the potential bias introduced by unequal variances, we performed Welch’s t-tests for those rsfMRI measures with significant F-tests and then correlated the Welch’s t-statistics with Student’s t-statistics. We then correlated the p-values from the two tests as well.” (lines 210-216)

14. Please report on model assumptions for multiple linear regression and how any violations were handled.

[Response] There are several model assumptions for multiple linear regression which we tested: no multicollinearity between predictor variables, no autocorrelation between the residuals of the data, homoscedasticity of the residuals, and the normality of the residuals. 

The following lines were added to the methods section of our manuscript:

“Several assumptions for multiple linear regression were assessed. Multicollinearity was assessed with the variance inflation factor (VIF) using the vif() function in R from the car package. Auto-correlation of the model residuals was assessed with the Durbin-Watson test using the durbinWatsonTest() function in R from the car package. Homoskedasticity of model residuals were assessed using the Breusch-Pagan (BP) test (using the ols_test_breusch_pagan() function in R from the olsrr package) and also visually inspected by plotting model residuals against marginal model fitted-values. For regressions with significant BP tests, we also performed weighted-least-squares (WLS) regression, which is able to account for differences in variance in the residuals, and then correlated the t-statistics from the WLS and the original ordinary-least squares (OLS) regressions to assess the impact of potential heteroscedasticity on the results. Finally, we performed a visual inspection of density plots of the residuals from regressions for 3 representative rsfMRI measures for each set of linear regression analyses, and their Quantile-Quantile (QQ) plots to detect any patterns of non-normality.” (lines 261-275)

Additionally, we have now included these lines in the results section of our manuscript:

“Several assumptions for multiple linear regression were assessed across all the analyses. Multicollinearity between predictors was assessed using the variance inflation factor (VIF) where GVIF values greater than 5 indicate significant multicollinearity between predictors in a multiple linear regression model. For the regressions performed across the ABCD 1.0, ABCD 4.0 (excluding 1.0), and full ABCD 4.0 samples, we obtained GVIF values for predictors in each regression and then found their average values across all regressions (Supplementary Table. F and G). No regressions exhibited significant multicollinearity between predictors. We performed Durbin-Watson tests to detect auto-correlation between the residuals from each of the regressions and found all test statistics were between 1.5 - 2.5, and thus within the acceptable range for auto-correlations (Supplementary Tables. 4-9). 

We performed the Breusch-Pagan (BP) test to the assumption of homoscedasticity for the model residuals from all the regressions (Supplementary Tables. 4-9). For the regressions using the ABCD 1.0 sample, one rsfMRI measure associated with anhedonia exhibited a significant BP test. For the regressions using the ABCD 4.0 (excluding 1.0) and full ABCD 4.0 samples, 4 rsfMRI measures associated with anhedonia exhibited significant BP tests, each. For the rsfMRI measures with significant BP tests, we plotted their model residuals against marginal model fitted values and did not detect any significant patterns of heteroskedasticity, visually (Supplementary Figures. 6-11 (left)). To be sure, we performed weighted-least-squares (WLS) regressions, which is able to account for differences in variance in the residuals, for these rsfMIR measures, and then correlated the t-statistics from the WLS and the original ordinary-least squares (OLS) regressions. The correlations were all equal to 1, indicating little effect of potential unequal variances on the results (Supplementary Figures. 6-11 (right)). 

Finally, we performed a visual inspection of density plots of the residuals from regressions for 3 representative rsfMRI measures significantly associated with anhedonia and their qq-plots for each set of regressions across the ABCD samples. All appeared normally distributed (Supplementary Figures 12-17).” (lines 649-677)

15. How many families were in the study? ICC values for the nesting factor?

[Response] For the ABCD 1.0 sample, there were 2163 families. For the ABCD 4.0 (excluding ABCD 1.0) sample, there were 5754 families. For the full ABCD 4.0 sample, there were 7688 families. We have now included the ICC values for each of the mixed model regressions we performed in the corresponding supplementary tables (Supplementary Tables. 4-9 ). We have included the following lines in our manuscript:

“We note that there were 2163 families in the ABCD 1.0 sample, 5754 families in the ABCD 4.0 (excluding 1.0) sample, and 7688 families in the ABCD 4.0 sample. The average Intra-Class-Correlation (ICC) across the regressions using the ABCD 1.0 sample was 0.09 (sd = 0.07) indicating that the random effects of family structure accounted for about 9% of the variability in rsfMRI connectivity measures (Supplementary Tables. 4 and 5). However, the standard deviation was fairly large, indicating significant variability across regressions. Similarly, for regressions using the ABCD 4.0 (excluding 1.0) sample and the full ABCD 4.0 sample, the average ICC across regressions was 0.10 (SD = 0.05) and ICC was 0.11 (SD = 0.05), respectively (Supplementary Tables. 6-9).” (lines 633-642) 

Results

16. Line 237--what findings?

[Response] We agree the sentence was not clear and have edited the manuscript which now reads:

“In line with the previous authors’ findings, we reproduced significant differences in 11 rsfMRI connectivity measures between those with and without anhedonia, indicated by the lnBF(10) statistic, though the effect sizes were small (Table. 2, left)(13).” (lines 309-311)

17. Lines 247-249: only 4 connectivity measures survived correction for multiple comparisons. How does this map onto the findings of the original study after correcting for multiple comparisons?

[Response] For our t-tests, we provided fairly conservative Benjamini-Hochberg (BH) adjusted p-values as we corrected for 306 comparisons (since there were 306 rsfMRI measures in total). The previous authors reported similar adjusted p-values as well as the log(Bayes Factor) (lnBF) statistic, which is a less conservative approach to assessing statistical significance. As such, although only 4 connectivity measures appeared to survive the more conservative BH adjusted p-value threshold, all 11 measures exhibited lnBF values > 1.1, which was interpreted as indicating significant evidence for the alternative hypothesis of there being significant mean differences between the anhedonia and control groups. We have clarified this in the results section of the manuscript which now reads:

“In addition to the lnBF statistic, we provide a more conservative adjustment for multiple comparisons using the Benjamini-Hochberg correction for 306 comparisons. Alternatively, since we were predominantly interested in reproducing the 11 rsfMRI associations reported by the previous authors, we could have adjusted for only 11 comparisons and arrived at a more liberal adjusted p-value for each comparison. However, we decided to report the former since we did indeed perform 306 t-tests in our analyses. This reflects the somewhat arbitrary nature of statistical thresholding as the number of outcome measures considered in family-wise hypotheses can be difficult to clearly define (22).” (lines 324-332)

18. Tables 2-4 are cut off. Do the tables include effect sizes and confidence intervals? If not, please include.

[Response] We have made significant revisions to our tables and now include effect sizes and their confidence intervals. The effect sizes we now report are Cohen’s d for t-tests (Table. 2) and the proportion (%) of the total variance in each rsfMRI measure accounted for by each independent variable (ex. anhedonia) in the multiple linear regressions (Tables. 5 and 6). 

19. Given the number of comparisons made and the sample size, the risk of false positives is high, and only findings that survive multiple comparisons should be interpreted.

[Response] We agree with the reviewers that the risk of false positives is high due to the large number of comparisons made. However, we have shown that the most statistically significant findings, such as those that survive multiple comparisons corrections, may paradoxically lead to focusing on the most inflated and thus least replicable findings if a sample is not well-powered enough. This is a major barrier in replication studies and we have included the following lines in the discussion to elaborate on this issue. 

“Using t-tests, only the Within-Cingulo-Opercular rsfMRI measure was consistently associated with anhedonia across the ABCD 4.0 (excluding 1.0) and full ABCD 4.0 samples. However, when we controlled for demographic covariates (sex, age, and race/ethnicity) using a linear regression approach, the association was no longer replicable in the ABCD 4.0 (excluding 1.0) sample. Like the other associations identified by the previous authors, we observed a significant decrease in effect size in the replication analysis after controlling for these additional covariates suggesting the presence of significant confounding effects that were not accounted for with t-tests. Furthermore, large decreases in effect size in replication analyses have been reported to occur more frequently when the initial discovery sample is small such that the most inflated and statistically significant findings are the most likely to be identified and reported (26). Unfortunately, these inflated findings are the least replicable as regression towards the mean leads to reductions in both effect size and significance in subsequent replications. Of note, Marek et al., 2022 found that controlling for sociodemographic covariates generally reduced effect sizes, and thus may help reduce effect size inflation.” (lines 682-697)

We found that when we included rsfMRI measures associated with anhedonia at the nominal p-value level (but not at the adjusted p-value level) in our multiple linear regressions in the ABCD 1.0 sample, we were able to identify two new rsfMRI measures associated with anhedonia which, although exhibiting smaller effect sizes, were replicable in the ABCD 4.0 (excluding 1.0) sample. We have included the following lines in the discussion regarding this point:

“Notably, these 2 rsfMRI measures exhibited smaller, and less statistically significant, effect sizes that remained relatively consistent across the regressions using the ABCD 4.0 (excluding 1.0) and full ABCD 4.0 samples. Thus, these findings suggest that replicability may be improved if less emphasis was placed on associations with the most statistical significance but rather on identifying those with the most stable effects across analyses, even if they are less statistically significant.” (lines 702-708) 

20. The authors make a number of comments about statistical power. Given the sample size, there is sufficient power to detect significant effects if they exist, and in fact the risk of false positives is increased in this larger sample. Rather, the more informative measure in such a large sample would be the magnitude of that effect. Effect sizes should be reported and interpreted for all findings, as well as confidence intervals.

[Response] We agree with the reviewer and have now included effect sizes and confidence intervals for all our analyses (see response to reviewer 1 comment 18). 

21. In table 4, is 'Effect of Anhedonia' supposed to be the B value?

[Response] Yes, in our original table, the “effect of anhedonia” is the partial regression coefficient (beta value). We have edited our tables significantly and now all our regression results are presented in Tables. 5 and 6. We have included the following lines in the table legends to clarify the values that are presented:

 “Effect size represents the percentage of the total variance (proportion of variance * 100%) in a rsfMRI connectivity measure accounted for by anhedonia. The partial regression coefficient (Estimate), standard error (Std.Err), nominal p-value (p), and conservative Benjamini-Hochberg adjusted p-value (p.adj) for anhedonia are also shown for each rsfMRI regression model. Bolded results were significant at the adjusted p-value level (p.adj < 0.05).” (lines 549-552)

22. R2 values are quite small--effects are extremely small. Authors should note this as a limitation--effect does not seem to be meaningful.

[Response] We agree with the reviewer’s comments and have added the following lines to the results section of our manuscript:

“We note that the effect sizes were extremely small, with anhedonia accounting for less than 1% of the total variance in each rsfMRI measure.” 

We also added the following lines to the discussion: 

“Although we found 2 rsfMRI measures with replicable associations, the actual effect sizes were extremely small, with anhedonia accounting for about 0.2% of the total variance for each rsfMRI measure. By themselves, these findings are unlikely to be clinically meaningfull. However, combining the small effects of many brain-based measures together may produce signals with significant clinical utility in applications such as in diagnostic prediction or the monitoring of disease progression (27,28). An analogous approach has been taken in genetic research whereby many genetic variants, which individually exhibit miniscule amounts of association, can be combined to produce genetic risk scores (29) that altogether contribute meaningfully to the prediction of the course of complex neuropsychiatric disorders (30).” (lines 717-726)

23. What was the n for the multiple regression analyses? I don't think a sample of thousands can be considered 'relatively small' as stated in lines 378 and 447.

[Response] For regressions using the ABCD 1.0 sample, the total n was 2424 with 215 individuals with anhedonia and 2209 controls. For regressions in the ABCD 4.0 (excluding 1.0) sample, the total n was 6454 with 591 individuals with anhedonia and 5863 controls. For regressions in the full ABCD 4.0 sample, the total n was 8864 with 800 individuals with anhedonia and 8864 controls. The n for the multiple regression analyses were reported in Tables. 1, 3 and 4. They have now also been included in Tables. 5 and 6. 

We have removed lines 378 and 447 from the manuscript. We agree with the reviewers that generally speaking, the ABCD study sample size is quite large. However, we have shown that the effect sizes for anhedonia are much smaller than expected and that the sample size from the ABCD 1.0 release may not have been large enough to detect associations that were significant and replicable with larger samples, especially with the inclusion of additional covariates in the regression models. We have included the following lines to the results section to elaborate on this point:

“We note that the Auditory vs. Right Putamen and Retrosplenial-Temporal vs. Right-Thalamus-Proper measures were not found to be associated with anhedonia in the ABCD 1.0 sample (Table. 6, left) at the adjusted or nominal p-value levels (although there was a trend). However, we see that the effect sizes and partial regression coefficients were similar to those estimated in the corresponding regressions using the ABCD 4.0 (excluding 1.0) and full ABCD 4.0 samples, albeit with wider confidence intervals and larger standard errors. These patterns suggest that the ABCD 1.0 sample size was not well-powered enough, after including the additional psychiatric comorbidity measures, to detect these associations.” (lines 612-620) 

Discussion

24. The meaning of the networks identified are not really elaborated on in the discussion, nor are clinical implications at least speculated. I might recommend the authors putting the knowledge expanded on in this study into clinical context.

[Response] We have now included the following lines in the discussion to elaborate on the meaning of the networks identified and clinical implications:

“The presence of anhedonia was associated with a decrease in the Auditory vs. Right-Putamen connectivity measure. Previous functional neuroimaging studies have found that functional activation between regions of the auditory cortex and putamen occurs during speech learning (39). Specifically, higher coactivation was associated with incorrect categorization of auditory stimuli. The authors proposed that the putamen may act to “tune” activity in the auditory cortices to help facilitate learning how to correctly categorize tones that lead to positive feedback. In the context of anhedonia, reduced connectivity between auditory and the putamen may reflect a more general impairment in reward based learning (40). For example, individuals with high levels of anhedonia have been shown to exhibit diminished ability to learn to modify their behavior during certain tasks in order to obtain larger rewards (41). Decreased connectivity between the striatal reward regions, such as the putamen, and cortical networks may disrupt the processes that drive behavioral and motivational adaptations that, in part, characterize anhedonia. , allowing us to disentangle rsfMRI connectivity measures associated with more than one condition based on t-test results. 

Anhedonia was also associated with decreases in the Retrosplenial-Temporal vs. Right-Thalamus-Proper connectivity measure. Few studies have reported on this connectivity measure in the context of anhedonia. The retrosplenial cortex has been associated with many cognitive processes, notably with impaired memory (42). Interestingly, one study showed that individuals with social anhedonia exhibited increased functional connectivity between the retrosplenial cortex and several other brain regions, which were also correlated with lower anticipated pleasure in social situations that may point to the role of the retrosplenial cortex in future-oriented thinking (43). As the thalamus has important functions in emotion and arousal (44), we speculate that a disrupted connectivity between retrosplenial networks and the thalamus may be associated with impaired future-oriented thinking about emotionally rewarding things or situations that may underlie the decreased motivational aspects of anhedonia. Interestingly, a recent study found significant anatomical connectivity between the restrosplenial cortex and fronto-striatal reward regions, suggesting a more direct involvement of the retrosplenial cortex in reward and decision making processes (45).

Altogether, our findings suggest that disrupted brain connectivity associated with anhedonia may underlie impairments in learning, emotional, and motivational processes. If brain dysfunction is a causal factor in the development of these dysregulated processes, then these brain processes may become significant targets of preventative measures and therapeutic treatments, including pharmacological or psychotherapeutic interventions. ” (lines 770-804)

Minor:

- One citation in the introduction (line 68) is in a different format compared to all other citations

[Response] The citation has been edited to match the format of the other citations. 

- 'prognostic predictor' in line 69 of the introduction is redundant

[Response] The word “prognostic” has been removed. 

- In line 128, ROIs should be plural, not possessive (ROI's)

[Response] “ROI’s” has been changed to “ROIs”. 

- Line 225, the two groups don't' appear to exhibit', they DO statistically exhibit differences.

[Response] The manuscript has been edited and now reads: 

“Note that although the two groups exhibit differences in a few of these characteristics, they were not controlled for statistically when we performed our t-tests.” (lines 294-296)

Reviewer #2

1. My main concern is that I didn’t feel like I got a good understanding of what was and what was not reproduced/replicated. Part of the issue is that the results were worded much differently in the original paper, with a focus on directionality of differences. I would very much like to see a summary table that provides a summary of sets of findings directly compared. It’s an arduous process to go through and double check that indeed all differences are reproduced/replicated. I feel particularly strong about this point in that statistical thresholding is always to some extent arbitrary – seeing results with p values displayed in parallel may help illustrate what does and what does not replicate.

[Response] We have now created a summary table (Table. 2) comparing the t-test results from across the reproduction/replication sets of analyses for easier interpretation. We also acknowledge the arbitrariness of statistical thresholding and discuss the 3 types (p-value, p.adjusted, and lnBF values) of statistical thresholding values reported. We have added the following lines to the manuscript to elaborate on this point:

“In addition to the lnBF statistic, we also provide a more conservative adjustment for multiple comparisons using the Benjamini-Hochberg correction for 306 comparisons. However, since we were predominantly interested in reproducing the 11 rsfMRI associations reported by the previous authors, we could have alternatively adjusted for only 11 comparisons and arrived at a more liberal adjusted p-value for each comparison. However, we decided to report the former since we did indeed perform 306 t-tests in our analyses. This reflects the somewhat arbitrary nature of statistical thresholding as the number of outcome measures considered in family-wise hypotheses is not always easily defined (22).” (lines 324-332)

Similarly, we have created summary tables comparing our results from multiple linear regressions across the ABCD samples (Tables. 5 and 6) in order to more easily interpret which rsfMRI measures were replicated. Importantly, we have included effect sizes for all our analyses to provide a clearer picture of the meaning of the results. 

2. It was not clear to me that all analyses from the original rsfMRI paper were reproduced. What about the analyses showing specificity?

[Response] The previous authors performed separate t-tests for youth-reported depressed mood and anxiety, and parent-reported ADHD (in the child) to assess the specificity of their findings. The exact reason for choosing these 3 psychiatric conditions was not clear. Nevertheless, we were able to reproduce the t-test results for anxiety and depressed mood symptoms (Supplementary Figure. 1)(See response to reviewer 1, comment #7 for more details). As of the ABCD 4.0 data release, the ABCD consortia released a notice stating diagnostic criteria for ADHD, among other disorders, required revision and that the corrected diagnostic data would not be available at this time. As such, we did not reproduce the previous authors’ findings for ADHD. 

The previous authors then performed follow-up t-tests directly comparing participants with anhedonia with those with depressed mood, anxiety, and ADHD separately to further investigate specificity. For the follow-up comparisons, participants reporting between comorbidity between anhedonia and one of the other psychiatric conditions were excluded from their analyses. After careful consideration, altogether, we do not believe this approach is appropriate for several reasons:

1. Excluding participants with anhedonia and a co-morbid condition(s) would greatly reduce the sample size for these comparisons, further reducing the statistical power necessary to detect what we now know are very small effects. Thus, findings are more likely to be null or significantly inflated. In fact, the majority of the previous authors’ follow-up tests were null findings, suggesting a general lack of specificity or lack of power. 

2. Many of the results from these follow-up t-tests are difficult to interpret in the context of the results of the more general t-tests. For example, the previous authors found that the CinguloParietalRightPallidum measure was significantly associated with anhedonia but not depressed mood from the general t-tests comparing anhedonia to controls, and depressed-mood to controls, suggesting specificity for anhedonia. However, their direct comparisons between those with anhedonia and those with depressed-mood showed no significant differences, suggesting lack of specificity, contradictory to the previous results. Again, lack of statistical power due to reduced sample sizes could have contributed to these null findings. 

3. Separate t-tests for anhedonia and other psychiatric conditions cannot tell us about the independent contributions these conditions have on the rsfMRI measures of interest. For example, the previous authors found the WithinRetrosplenialTemporal measure to be independently associated with anhedonia and depressed mood, suggesting lack of specificity. When they compared those with anhedonia to those with depressed-mood directly, they also found no difference further suggesting lack of specificity. However, we have no information on to what extent anhedonia or depressed mood independently contribute to the rsfMRI measure. 

For our specificity analyses, we decided to focus on 3 psychiatric conditions found to be significantly correlated with anhedonia, which were symptoms of depressed mood and irritability, and bipolar II disorder (Figure. 1). Like the previous authors, we performed separate t-tests for each of the co-morbid conditions and then assessed whether the rsfMRI measures associated with anhedonia were also associated with the other conditions. We did not perform follow-up t-test analyses directly comparing those with anhedonia to those with the other psychiatric conditions for the reasons mentioned above. Specificity was further explored with our multiple linear regression analyses. We highlight the following lines added to the results section of our manuscript:

“In line with the previous authors’ approach, in order to assess the specificity of the associations found for anhedonia, we next performed t-tests to compare rsfMRI connectivity measures separately between those with and without symptoms of depressed mood, irritability, and bipolar II disorder using the full ABCD 4.0 sample. The presence of a psychiatric condition was the reference group. 

We found that 2 rsfMRI measures significantly associated with anhedonia using the full ABCD 4.0 sample were also significantly associated with depressed mood. These were the Default vs. Dorsal-Attention (Cohen’s d = -0.118, 95%CI [-0.191, -0.045], lnBF = 1.807) and Within-Cingulo-Opercular (Cohen’s d = 0.117, 95%CI [0.043, 0.19], lnBF = 1.646) connectivity networks (Supplementary Table. 13). Similarly, 2 rsfMRI measures associated with anhedonia were also significantly associated with irritability. These were the Salience vs. Left-Ventraldc (Cohen’ d = -0.139, 95%CI [-0.224, -0.054], lnBF = 2.075) and Default vs. Dorsal-Attention (Cohen’s d = -0.138, 95% CI [-0.223, -0.053], lnBF = 2.004) connectivity measures (Supplementary Table. 13). None of the rsfMRI measures associated with anhedonia were also associated with bipolar II.” (lines 479-494)

3. I find it curious that the paper seems to suggest that the analyses were run by a team of independent researchers when the first author of the initial paper is the second author of the current paper if I'm not mistaken. It’s not an issue per se, but I want to make sure that all coding for the reproducibility analyses was done from scratch, otherwise it may just carry forward errors.

[Response] The middle author of this manuscript (Narun Pat) is indeed the first author of the initial paper whose results we are trying to replicate in this manuscript. At the start of this project, the primary author of this paper (Yi Zhou) was not able to exactly reproduce the results from the previous paper. Upon reaching out to Narun, he provided the R code used for the original paper (which was not publicly available) and it was discovered that participants with missing responses for anhedonia were removed prior to the identification and exclusion of outliers for rsfMRI connectivity measures (outliers defined as data points exceeding 1.5 times the IQR). This small arbitrary discrepancy in the order of data processing steps prevented the exact reproduction of the previous results. Since Narun had been kind enough to share his code, as well as aid in the review of the manuscript before its submission, we invited him to be a co-author.

We acknowledge that while we used his code to guide our reproduction and replication analyses, the code for this current manuscript was otherwise written from scratch. Furthermore, we were critical of many of the analytical decisions made in the original paper (ex. not controlling for covariates, not including covariates during corrections for batch effects, and problematic specificity analyses, etc) and subsequently applied alternative ones in our second set of analyses utilizing a multiple linear regression approach. Currently, it is common practice for analysis code to be made publicly available alongside a published paper in order to increase the transparency of analytical decision making that ultimately contributes to greater reproducibility/replicability efforts. 

4. May be worth presenting the correlations between covariates and anhedonia in the supplements. How were covariates decided on? Why was trauma or anxiety not included for example?

[Response] Previously, we had included the correlation matrix between youth-reported anhedonia and 33 other psychiatric symptoms and diagnoses as a supplementary figure. We have now moved that figure into the main manuscript as Figure. 1. We identified depressed-mood, irritability, and bipolar disorder as significantly comorbid psychiatric measures considered in subsequent analyses because they were significantly correlated with anhedonia and exhibited correlation coefficient values greater than 0.5. While anxiety also appeared somewhat correlated with anhedonia, we limited our selection to those exceeding correlations of 0.5 as including too many covariates in our multiple linear regressions would significantly reduce our statistical power. We have revised the results section of our manuscript to clarify this point:

 “We took a step-wise approach and first only included the socio-demographic covariates (sex, age, race/ethnicity) in our regression models along with youth-reported anhedonia. Then, we added the comorbid psychiatric conditions (depressed mood, irritability, and bipolar II) to the models in order to evaluate their impact on the regression estimates and the replicability of any significant associations with anhedonia. We limited the inclusion of comorbid psychiatric conditions to these three measures in order to preserve the statistical power of our regression analyses and to account for comorbidities that are more likely to exhibit potential confounding effects based on their relatively higher correlations with anhedonia (Supplementary Table. 11).” (lines 522-530)

We also acknowledge the potential impact of environmental factors, such as trauma, on the associations between anhedonia and rsfMRI measures. These important considerations are unfortunately out of the scope of this manuscript. However, we have included the following lines in the limitations section of the discussion to address this point:

“For example, several recent studies have found that racial discrimination is associated with lower total brain volume (35) and alterations in prefrontal white matter tracts in adults (36,37). While out of the scope of this study, it will be critical to investigate how social determinants of health and other environmental factors, such as trauma (4,38), contribute to differences in health and brain-based outcomes between different racial and ethnic groups during child and adolescent development.” (lines 752-758)

5. Partial regression coefficients of all significant predictors should be displayed, not just the rsFMRI measures.

[Response] We have included the full multiple linear regression results across the different ABCD samples in our supplementary tables which includes the partial regression coefficients for all predictors and their effect sizes (the %variance in the rsfMRI measure accounted for by each predictor) (Supplementary Tables. 4-9). Unfortunately, including all the significant predictors on one table would be difficult and also likely to limit interpretability of the findings. As such, we focused on the partial regression coefficients for anhedonia in our main results (Tables. 5 and 6). 

However, for the 2 rsfMRI measures with replicable effects and significant associations with anhedonia, we include a variable importance figure to visually compare the relative effect sizes of all the other predictors in the regression models (Figure. 2). The following lines were added to the discussion to elaborate on these findings:

“By including the other predictors, the multiple linear regression models accounted for about 4.5% of the total variance in the Auditory vs. Right Putamen rsfMRI measure and about 3% of the total variance for the Retrosplenial-Temporal vs. Right-Thalmaus-Proper rsfMRI measure (Figure. 2). For both regression models, the race/ethnicity predictor accounted for the vast majority of the explained variance. Upon inspection of the partial regression coefficients for each model, Black, Hispanic, and Other race/ethnicity exhibited the largest and most significant partial regression coefficients for both rsfMRI measures (Supplementary table. 9).” (lines 727-734)

6. The results from the new specificity analyses were not nearly discussed enough in the discussion section nor was the effect of controlling for additional covariates on the 1.0 sample.

[Response] We have made significant revisions to our previous specificity analyses. For the T-tests, we focused on the 11 rsfMRI measures previously found to be associated with anhedonia. Notably, 6 of the 11 rsfMRI measures were still associated with anhedonia in the full ABCD 4.0 sample. We performed separate sets of t-tests to compare rsfMRI connectivity measures separately between those with and without symptoms of depressed mood, irritability, and bipolar II disorder using the full ABCD 4.0 sample. We compared and identified which of the 11 rsfMRI measures were also associated with these 3 comorbid psychiatric conditions. For the t-test results, the following lines are now included in the results section:

“We found that 2 rsfMRI measures significantly associated with anhedonia using the full ABCD 4.0 sample were also significantly associated with depressed mood. These were the Default vs. Dorsal-Attention (Cohen’s d = -0.118, 95%CI [-0.191, -0.045], lnBF = 1.807) and Within-Cingulo-Opercular (Cohen’s d = 0.117, 95%CI [0.043, 0.19], lnBF = 1.646) connectivity networks (Supplementary Table. 13). Similarly, 2 rsfMRI measures associated with anhedonia were also significantly associated with irritability. These were the Salience vs. Left-Ventraldc (Cohen’ d = -0.139, 95%CI [-0.224, -0.054], lnBF = 2.075) and Default vs. Dorsal-Attention (Cohen’s d = -0.138, 95% CI [-0.223, -0.053], lnBF = 2.004) connectivity measures (Supplementary Table. E.1). None of the rsfMRI measures associated with anhedonia were also associated with bipolar II.

Altogether, the results of the t-test comparisons using the full ABCD 4.0 sample suggest the Cingulo-Opercular vs. BrainStem, Retrosplenial-Temporal vs. Right-Cerebellum-Cortex, and the Sensorimotor-Hand vs. BrainStem connectivity measures may be more specifically associated with anhedonia. Contrastingly, the Within-Cingulo-Opercular connectivity measure, whose association with anhedonia was the only one replicated at the more conservative adjusted p-value and lnBF statistic levels, was associated with both depressed mood and irritability.” (lines 484-501)

Additionally, we added the following lines in the discussion to elaborate on the impact of including sociodemographic covariates in the multiple linear regressions:

“Using t-tests, only the Within-Cingulo-Opercular rsfMRI measure was consistently associated with anhedonia across the ABCD 4.0 (excluding 1.0) and full ABCD 4.0 samples. However, when we controlled for demographic covariates (sex, age, and race/ethnicity) using a linear regression approach, the association was no longer replicable in the ABCD 4.0 (excluding 1.0) sample. Like the other associations identified by the previous authors, we observed a significant decrease in effect size in the replication analysis after controlling for these additional covariates suggesting the presence of significant confounding effects that were not accounted for with t-tests.” (lines 682-690)

For the multiple linear regressions, we assessed specificity by including the 3 comorbid psychiatric conditions as additional predictors in the regression models. We then assessed whether the partial regression coefficients for these additional comorbidities were also significantly associated with rsfMRI measures that were found to be associated with anhedonia. The following lines are now included in the results section:

 “Next, we added depressed-mood, irritability, and bipolar II disorder measures to the regression analyses across the ABCD samples in order to assess the impact of accounting for comorbid conditions on the specificity and stability of the effect sizes and significance of the associations between rsfMRI measures and anhedonia that were previously identified. 

For the Auditory vs. Right-Putamen and Retrosplenial-Temporal vs. Right-Thalamus-Proper measures, we found that their significant associations with anhedonia were preserved in the regressions using the ABCD 4.0 (excluding 1.0) and full ABCD 4.0 samples, even after accounting for psychiatric comorbidities (Table. 6, center and right). Furthermore, their effect sizes remained relatively consistent across the analyses using the different ABCD samples as well. Importantly, depressed-mood, irritability, and bipolar II disorder were not found to be significant predictors of these rsfMRI measures in any of the multiple linear regression analyses, suggesting these associations are specific to anhedonia (Supplementary Table. 17).” (lines 588-601) 

Furthermore, in the discussion, we elaborate on the meaning of the rsfMRI measures found to be specifically associated with anhedonia, as well as some potential clinical implications (see response to reviewer 1, comment 24).

7. The result section in the abstract should explicitly state that the larger full sample that replicated 6/11 associations included the data from the initial study and is therefore not an independent replication. In the same vein, the wording “replication of previous findings was limited” maybe overly optimistic given that in the independent sample only 1/11 was replicated, and, once controlled for covariates, a lot of the associations in the initial sample were no longer significant, too.

[Response] We have revised the results section of the manuscript which now reads:

“To increase our power to detect genuine associations with smaller effect sizes, we next performed our analyses using the full ABCD 4.0 release sample, including all participants from the ABCD 1.0 release. Since we are including the participants used in the initial analyses, our analyses using the full ABCD 4.0 sample would not be an independent replication of the previous results. Nevertheless, the results will help with the assessment of the stability of the effect sizes and associations.” (lines 409-414)

Our main findings were that associations between anhedonia and 2 rsfMRI measures (which were not part of the 11 measures identified by the previous authors) were replicable in the ABCD 4.0 (excluding 1.0) sample (which can be considered similar to an independent sample), even after accounting for demographic and psychiatric comorbidities. For these two rsfMRI measures, while statistical significance was not achieved in the ABCD 1.0 sample, this was likely due to the lower statistical power in this smaller sample. Notably, the effect sizes were consistent, and statistically significant, in the larger ABCD 4.0 (excluding 1.0) and full ABCD 4.0 samples, where there was greater statistical power. We have included the following lines in the results to clarify this point:

“We note that the Auditory vs. Right Putamen and Retrosplenial-Temporal vs. Right-Thalamus-Proper measures were not found to be associated with anhedonia in the ABCD 1.0 sample (Table. 6, left) at the adjusted or nominal p-value levels (although there was a trend). However, we see that the effect sizes and partial regression coefficients were similar to those estimated in the corresponding regressions using the ABCD 4.0 (excluding 1.0) and full ABCD 4.0 samples, albeit with wider confidence intervals and larger standard errors. These patterns suggest that the ABCD 1.0 sample size was not well-powered enough, after including the additional psychiatric comorbidity measures, to detect these associations.” (lines 612-620)

Finally, we have revised the abstract which now reads:

“Results: While the previously reported associations were reproducible, effect sizes for most rsfMRI measures were drastically reduced in replication analyses (including both t-tests and multiple linear regressions) using the ABCD 4.0 (excluding 1.0) sample. However, 2 new rsfMRI measures (the Auditory vs. Right Putamen and the Retrosplenial-Temporal vs. Right-Thalamus-Proper measures) exhibited replicable associations with anhedonia and stable, albeit small, effect sizes across the ABCD samples, even after accounting for demographic covariates and comorbid psychiatric conditions using a multiple linear regression approach. 

Conclusion: The most statistically significant associations between anhedonia and rsfMRI connectivity measures found in the ABCD 1.0 sample tended to be non-replicable and inflated. Contrastingly, replicable associations exhibited smaller effects with less statistical significance and multiple linear regressions helped assess the specificity of these findings and control the effects of confounding covariates.” (lines 34-47).

References 

1. Jakobsen JC, Gluud C, Wetterslev J, Winkel P. When and how should multiple imputation be used for handling missing data in randomised clinical trials – a practical guide with flowcharts. BMC Med Res Methodol. 2017 Dec 6;17(1):162.

2. Ghasemi A, Zahediasl S. Normality Tests for Statistical Analysis: A Guide for Non-Statisticians. Int J Endocrinol Metab. 2012;10(2):486–9.

3. Yang K, Tu J, Chen T. Homoscedasticity: an overlooked critical assumption for linear regression. Gen Psychiatry. 2019 Oct 17;32(5):e100148.

---

## [Decision Letter · Decision Letter 1]

1 Mar 2023

Associations Between Resting State Functional Brain Connectivity and Childhood Anhedonia: A Reproduction and Replication Study

PONE-D-22-28899R1

Dear Dr. Zhou,

We’re pleased to inform you that your manuscript has been judged scientifically suitable for publication and will be formally accepted for publication once it meets all outstanding technical requirements.

Kind regards,

Melissa A Brotman, PhD

Academic Editor

PLOS ONE

Additional Editor Comments (optional):

Reviewers' comments:

Reviewer's Responses to Questions

**Comments to the Author**

1. If the authors have adequately addressed your comments raised in a previous round of review and you feel that this manuscript is now acceptable for publication, you may indicate that here to bypass the “Comments to the Author” section, enter your conflict of interest statement in the “Confidential to Editor” section, and submit your "Accept" recommendation.

Reviewer #1: All comments have been addressed

Reviewer #2: All comments have been addressed

2. Is the manuscript technically sound, and do the data support the conclusions?

Reviewer #1: Yes

Reviewer #2: Yes

3. Has the statistical analysis been performed appropriately and rigorously? 

Reviewer #1: Yes

Reviewer #2: Yes

4. Have the authors made all data underlying the findings in their manuscript fully available?

Reviewer #1: Yes

Reviewer #2: Yes

5. Is the manuscript presented in an intelligible fashion and written in standard English?

Reviewer #1: Yes

Reviewer #2: Yes

6. Review Comments to the Author

Reviewer #1: (No Response)

Reviewer #2: I have no further comments. The authors did a great job with the revision - this is an important paper.

7. PLOS authors have the option to publish the peer review history of their article (what does this mean?). If published, this will include your full peer review and any attached files.

Reviewer #1: **Yes: **Lana Ruvolo Grasser

Reviewer #2: No

---

## [Editor Report · Acceptance letter]

7 Mar 2023

PONE-D-22-28899R1 

Associations Between Resting State Functional Brain Connectivity and Childhood Anhedonia: A Reproduction and Replication Study 

Dear Dr. Zhou:

I'm pleased to inform you that your manuscript has been deemed suitable for publication in PLOS ONE. Congratulations! Your manuscript is now with our production department. 

Kind regards, 

on behalf of

Dr. Melissa A Brotman 

Academic Editor

PLOS ONE